METHODS

# GPMelt: A hierarchical Gaussian process framework to explore the dark meltome of thermal proteome profiling experiments

**Cecile Le Sueur**[1,2], **Magnus Rattray**[3‡*], **Mikhail Savitski**[1‡*]

**1** Genome Biology Unit, European Molecular Biology Laboratory, Heidelberg, Germany, **2** Department of Biology, ETH Zürich, Zürich, Switzerland, **3** Faculty of Biology, Medicine and Health, University of Manchester, Manchester, United Kingdom

‡These authors jointly supervised this work.
* magnus.rattray@manchester.ac.uk (MR); mikhail.savitski@embl.de (MS)

**Data Availability Statement:** There are no primary data in the paper; all data used to produce the results are available on Zenodo

## Abstract

Thermal proteome profiling (TPP) is a proteome wide technology that enables unbiased detection of protein drug interactions as well as changes in post-translational state of proteins between different biological conditions. Statistical analysis of temperature range TPP (TPP-TR) datasets relies on comparing protein melting curves, describing the amount of non-denatured proteins as a function of temperature, between different conditions (e.g. presence or absence of a drug). However, state-of-the-art models are restricted to sigmoidal melting behaviours while unconventional melting curves, representing up to 50% of TPP-TR datasets, have recently been shown to carry important biological information. We present a novel statistical framework, based on hierarchical Gaussian process models and named GPMelt, to make TPP-TR datasets analysis unbiased with respect to the melting profiles of proteins. GPMelt scales to multiple conditions, and extension of the model to deeper hierarchies (i.e. with additional sub-levels) allows to deal with complex TPP-TR protocols. Collectively, our statistical framework extends the analysis of TPP-TR datasets for both protein and peptide level melting curves, offering access to thousands of previously excluded melting curves and thus substantially increasing the coverage and the ability of TPP to uncover new biology.

## Author summary

Proteins interactions with other proteins, nucleic acids or metabolites, are key to all biological processes. Being able to detect these interactions is essential to understand biological systems. Thermal proteome profiling is a proteome-wide biological assay able to capture these interactions. It consists in analysing the effect of heat treatment on proteins. Indeed, proteins, under physiological conditions, are folded. This folding gets disrupted as the temperature increases. The way this unfolding happens, called the melting profile of the protein, informs on the interactions of proteins. For example, the interaction of a protein with another protein can increase (thermally stabilise) or decrease (thermally

(DOI:http://doi.org/10.5281/zenodo.12806563). All code written in support of this publication is available on a Gitlab repository at https://git.embl.de/grp-savitski/gpmelt.git.

**Funding:** This work was supported by the European Molecular Biology Laboratory (EMBL). C. L.S. was supported by a fellowship of the EMBL International PhD program. The funders had no role in study design, data collection and analysis, decision to publish, or preparation of the manuscript.

**Competing interests:** The authors have declared that no competing interests exist.

destabilise) the temperature at which this protein starts unfolding. In this work, we present a new statistical method, named GPMelt, to analyse these melting profiles. Notably, GPMelt allows to analyse any melting profiles, independently of their shapes. The proposed improvements over previously published methods allow to investigate more robustly the melting profiles of more proteins, hence increasing the ability of thermal proteome profiling assays to discover new protein interactions. We anticipate that these advancements will aid in unravelling complex biological phenomena.

## Introduction

Proteins play a central role in all biological processes by interacting with a variety of biomolecules, including other proteins, nucleic acids and metabolites. By modifying proteins' physico-chemical properties, these interactions impact proteins' thermal stability, which describes a protein's tendency to denature and aggregate under heat-treatment. Thermal stability can be monitored via Thermal Proteome Profiling (TPP) [1], a proteome-wide technology combining the principle of the cellular thermal shift assay [2] with quantitative mass spectrometry (MS) [3, 4]. The data analysis entails the construction and comparison of thousands of proteins' melting curves, describing the melting behavior of proteins. Initially developed to detect targets and off-targets of drugs [1], the versatility of TPP has been broadly extended since its development both in terms of technology [5–8] as well as biological applications [9–13].

In this work, we focus on Temperature Range TPP (TPP-TR) experiments, in which the soluble fractions of each heated sample are analysed by multiplexed quantitative mass spectrometry [4] to obtain relative quantification between the soluble proteins at different temperatures, thus determining their melting curves. Here, the quantitative MS measurements of all tryptic peptides corresponding to a single gene entry in the used proteome database are combined to obtain the protein-level melting curve. However, the consideration of peptide-level melting curves from these MS measurements is also possible and is especially of interest for the study of post translational modifications (PTMs) [11] and the detection of proteoforms [14]. Proteoforms encompass all possible molecular modifications (among which genetic variations, alternative splicing and PTMs) affecting the protein product of a single gene [15]. The quantification of proteins (or peptides) abundances at successively increasing temperatures allows the reconstruction of a melting curve for each protein (or peptide). Following the chemical denaturation theory [16], these melting curves are usually assumed to have a sigmoidal shape [1, 6]. Changes in thermal stability of a protein due to a treatment, e.g. addition of a drug, can be statistically assessed by finding significant shifts in melting curves between control and treatment conditions. Changes in melting behaviours of a protein due to e.g. PTMs can be similarly detected by investigating peptides melting curves.

However, as mentioned in recent works [17, 18] a non-negligible portion of proteins show non-sigmoidal melting behaviours. Sridharan et al. [19] suggested that these unconventional melting curves are likely to carry important biological information, as they can reveal complex temperature dependent dynamics of phase separated proteins [20, 21]. Phase separated proteins are proteins participating in macromolecular condensates, whose formations and/or dissolutions can be affected by temperature [22]. Furthermore, the study of peptide-level melting curves also involves a larger amount of unconventional melting behaviours. Indeed, protein-level melting curves are obtained by averaging melting behaviours over multiple peptides. This results in a smoother estimate of the melting behaviour, which tends towards a more sigmoidal shape. On the contrary, peptide-level melting curves, considered individually, suffer from a

larger replicate to replicate variability and a larger amount of unconventional melting behaviours.

In this work, we introduce a new statistical framework, named GPMelt, to fit and compare the melting curves obtained from TPP-TR experiments. This general method is shown to be applicable to both protein- and peptide-level observations. Moreover, our model presents several new features enabling us to analyse TPP-TR data with more sensitivity and robustness than previous methods. Firstly, the model does not rely on parametric functions to describe the melting curves. This means that proteins are no longer filtered out or incorrectly fitted due to deviations from a sigmoidal melting behaviour. Hence, this development substantially expands the meltome accessibility compared to the two state-of-the-art methods for TPP-TR dataset analysis, namely the melting point ($T_m$) approach [1, 8] and NPARC (Non-parametric Analysis of Response Curve) [6]. Furthermore, our model can handle multiple conditions at once, and is especially versatile, as deeper hierarchies extend it to more complex experimental setups. Deeper hierarchies are obtained by adding hierarchical levels to the model, with these additional levels helping to account for more complex correlation structures originating from more complex protocols. The parameters of the models are interpretable and can, in particular, be used to detect outliers in the dataset. Moreover, the model fit makes it possible to suggest a new measure of distance between curves, that we call the Area Between the Curves (ABC), to replace the previously used $\Delta T_m$, which is only valid under a sigmoidal assumption on the melting curves. Lastly, we introduce a new scaling of the observations, called the mean scaling. Unlike the widely used fold change scaling, the mean scaling doesn't force the melting curves to start at one, respects the statistical assumptions on the model and improves the reproducibility of replicates within each condition.

We start by introducing GPMelt on protein-level TPP-TR using a three-level hierarchical Gaussian process model. We further demonstrate how this model can easily be extended to incorporate multiple conditions. Finally, motivated by peptide-level TPP-TR dataset analysis, we expand the model specification to deeper hierarchies. GPMelt is subsequently validated using multiple published datasets, among which five are protein-level and one is a peptide-level TPP-TR dataset. A high-level visualisation the TPP-TR protocol and GPMelt framework is depicted in Fig 1.

## Methods

In this work, we propose to model melting curves via a hierarchical model based on Gaussian Processes (GPs), and to assess significance of differentially melting curves using a statistic, denoted hereafter by $\Lambda$, whose distribution under the null hypothesis has to be approximated. We first introduce the reader to GP regression, and how it links to previously developed methods to analyze TPP-TR data. We then build the proposed hierarchical model, and present our hypothesis testing framework. Next, we show how the model can deal with multiple conditions. Finally, we introduce deeper hierarchies to tackle more complex TPP-TR protocols and biological questions. A summary of the main algorithmic steps is represented in Fig A in S1 File.

### Notation

We start by introducing our model for protein-level TPP-TR datasets, along with some notation (see Table 1). Observations for protein $p$, with $p \in [\![P]\!]$, come from one or multiple conditions $c \in [\![C_p]\!]$, each condition having several replicates, denoted by $r \in [\![R_{pc}]\!]$. For simplicity, we assume these replicates to have been measured at the same set of $N$ temperatures $T = [t_1, \ldots, t_N]^T$ although the model implementation allows for asynchronous observations.

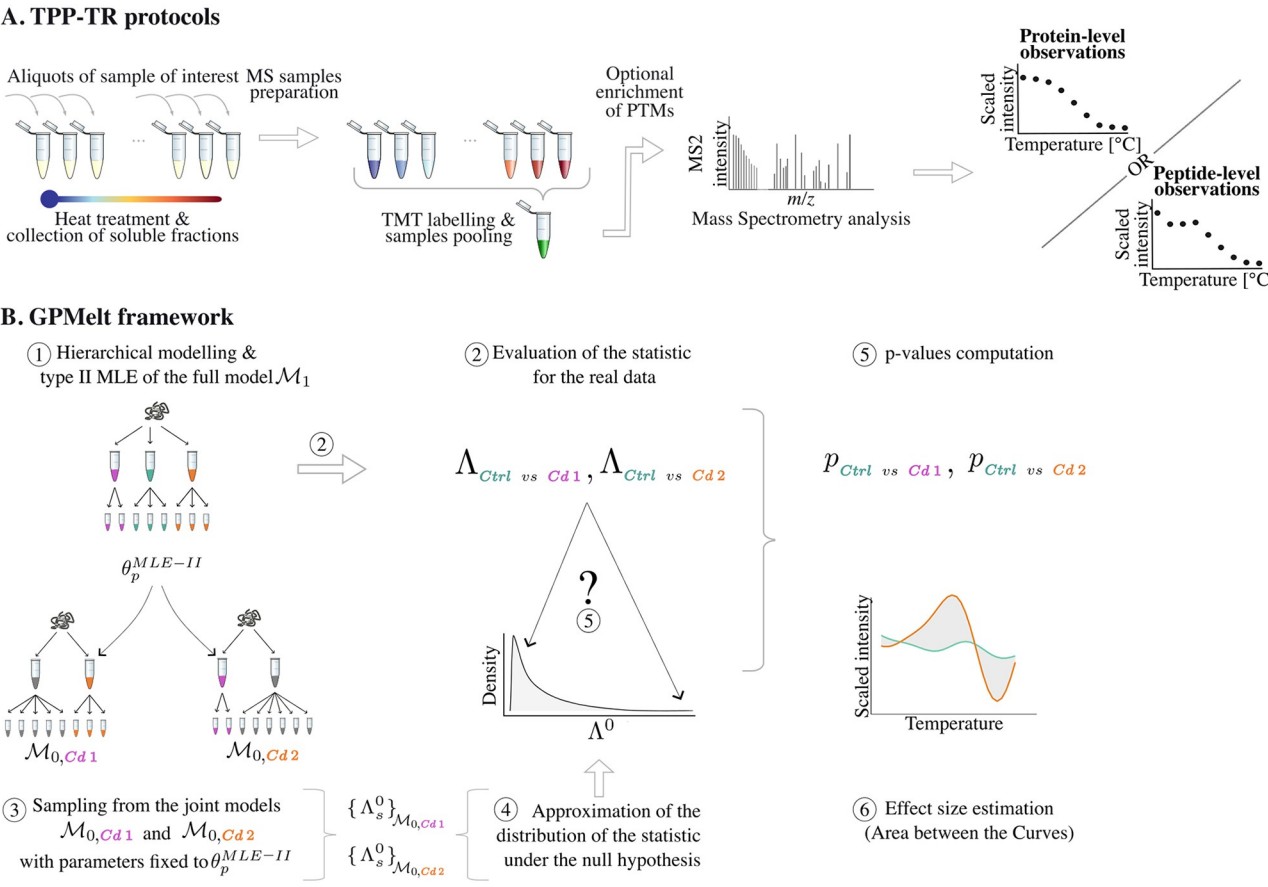

**Fig 1. A schematic visualisation of the TPP-TR assay and the GPMelt framework.** (A) Temperature-Range Thermal Proteome Profiling (TPP-TR) protocol: 10 aliquots of cells or cell lysates are heated to a range of temperatures for three minutes. Subsequently the formed aggregates are removed by filtration or centrifugation. The soluble fractions is digested with trypsin and labeled with tandem mass tags and then analyzed by mass spectrometry (MS). Acquired data, which are on the peptide-level, can be combined to obtain the protein-level melting curves, by averaging over the measurements of all tryptic peptides corresponding to a single protein entry in the protein database. Permission to use and modify the tubes and MS spectrum icons was kindly granted by Isabelle Becher. (B) The GPMelt framework consists in fitting simultaneously all replicates of all conditions independently for each protein, using a hierarchical Gaussian process (HGP) model (1). This model, denoted by $\mathcal{M}_1$, is called the *full* model. The full model is fitted via type II maximum likelihood estimation (type II MLE), and estimated parameters $\theta_p^{MLE-II}$ are plugged into the so-called *joint* models (3). A joint model $\mathcal{M}_0$ corresponds to a model in which at least two conditions are jointly modeled as one (these two conditions are represented in grey in $\mathcal{M}_{0,Cd1}$ and $\mathcal{M}_{0,Cd2}$). The joint models are used to generate an approximation of the null distribution of the GPMelt statistic $\Lambda$ (4). The observed statistics $\Lambda_{Ctrl\ vs\ Cd1}$ and $\Lambda_{Ctrl\ vs\ Cd2}$ (2) are compared to this null distribution approximation to compute empirical p-values (5). Additionally, an effect size is computed using the predicted fits obtained from $\mathcal{M}_1$ (6).

We develop a model of the scaled abundances $Y_{pcr}$ of protein $p$ in replicate $r$ of condition $c$, which have been observed at $N_{pcr}$ temperatures denoted by $T_{pcr}$, with $N_{pcr} \leq N$, $T_{pcr} = [t_1^{pcr}, \ldots, t_{N_{pcr}}^{pcr}]^T \subseteq T$ and $Y_{pcr} = [y_1^{pcr}, \ldots, y_{N_{pcr}}^{pcr}]^T$.

As a generalisation, we present the model without specifying the scaling factor. More precisely, if $\Gamma_{pcr} = [\gamma_1^{pcr}, \ldots, \gamma_{N_{pcr}}^{pcr}]^T$ are the raw abundances measured for replicate $r$ in condition $c$ of protein $p$, then the scaled observations $Y_{pcr}$ are obtained by

$$ y_i^{pcr} = \frac{\gamma_i^{pcr}}{\rho_{pcr}} \quad \forall \quad i \in [\![1, N_{pcr}]\!] \tag{1} $$

**Table 1. Notation.**

| Notation | Definition |
|---|---|
| $p \in [\![1, P]\!]$ | Set of proteins |
| $c \in [\![1, C_p]\!]$ | Set of conditions for protein $p$ |
| $r \in [\![1, R_{pc}]\!]$ | Set of replicates for condition $c$ of protein $p$ |
| $\pi_j$ with $j \in [\![1, \Pi_p]\!]$ | Set of observed peptides for protein $p$ |
| $N$ | Number of temperatures in the TPP-TR experiment |
| $N_{pcr} \leq N$ | Number of temperatures at which replicate $r$ of condition $c$ of protein $p$ has been observed |
| $N_{pc} = \sum_r N_{pcr}$ | Total dimension of the observations for condition $c$ of protein $p$ |
| $N_p = \sum_c N_{pc}$ | Total dimension of the observations for protein $p$ |
| $T = [t_1, \dots, t_N]^T$ | Set of temperatures measured during the TPP-TR experiment |
| $T_{pcr} = [t_1^{pcr}, \dots, t_{N_{pcr}}^{pcr}]^T \subseteq T$ | Set of temperatures at which replicate $r$ of condition $c$ of protein $p$ has been observed |
| $T_{pc} = [(T_{pc1})^T, \dots, (T_{pcR_{pc}})^T]^T \in \mathcal{R}^{N_{pc}}$ | Concatenation of all observed temperatures for all replicates in conditions $c$ of protein $p$ |
| $T_p = [(T_{p1})^T, \dots, (T_{pC_p})^T]^T \in \mathcal{R}^{N_p}$ | Concatenation of all observed temperatures for all conditions of protein $p$ |
| $\Gamma_{pcr} = [\gamma_1^{pcr}, \dots, \gamma_{N_{pcr}}^{pcr}]^T$ | Measured raw abundance for replicate $r$ of condition $c$ of protein $p$ |
| $\rho_{pcr}$ | Scaling factor for replicate $r$ of condition $c$ of protein $p$ |
| $y_i^{pcr} = \frac{\gamma_i^{pcr}}{\rho_{pcr}}$ | Scaled abundance at $t_i$ for replicate $r$ of condition $c$ of protein $p$ |
| $Y_{pcr} = [y_1^{pcr}, \dots, y_{N_{pcr}}^{pcr}]^T$ | Scaled abundances for replicate $r$ of condition $c$ of protein $p$ |
| $Y_{pc} = [(Y_{pc1})^T, \dots, (Y_{pcR_{pc}})^T]^T \in \mathcal{R}^{N_{pc}}$ | Concatenation of all scaled abundances of all replicates of condition $c$ of protein $p$ |
| $Y_p = [(Y_{p1})^T, \dots, (Y_{pC_p})^T]^T \in \mathcal{R}^{N_p}$ | Concatenation of all scaled abundances of all conditions of protein $p$ |
| $k_\kappa(t, t'|\lambda) = \sigma_\kappa^2 \cdot k(t, t'|\lambda)$ | Radial basis function kernel |
| $k(t, t'|\lambda) = \exp\left(-\frac{\|t - t'\|^2}{2\lambda^2}\right)$ | |
| $\theta_p$ | Set of all hyper-parameters of the hierarchical Gaussian process model for protein $p$ |
| $\theta_p^{MLE-II}$ | Type II maximum likelihood estimate of the set of hyper-parameters of the hierarchical Gaussian process model for protein $p$ |

with $\rho_{pcr}$ a scaling factor chosen to be constant across temperatures of each replicate. Scaling factors are discussed in more detail in the results section, in Fig 6E and 6F and in Supporting Information E in S1 File.

## Gaussian Process regression and its relationship to previous methods

In this section, we introduce Gaussian Process regression, and show how previously developed statistical methods [6, 17] for TPP-TR datasets can be seen as part of this framework. This section is summarized in Table B in S1 File.

The analysis of TPP-TR experiment is typically done protein-wise. We consider the aim of modelling the observations of one replicate of a condition of a protein $(T_{pcr}, Y_{pcr}) \in \mathcal{R}^{N_{pcr} \times 2}$, and simplify these notations to $(T, Y) \in \mathcal{R}^{N \times 2}$ in this section. Using this replicate, our aim is to infer the underlying melting curve describing the protein's melting behaviour in this condition. This melting curve can be modeled by a real-valued function $f$, which is only observed at a restricted set of temperatures $T$, and whose observations $Y$ contains measurement errors. We suggest to see this function $f$ as the realization of an unknown one-dimensional continuous

stochastic process. Here, the stochasticity models natural variations observed between biological replicates, distinguishable from batch effect and measurement errors.

A Gaussian Process is an example of continuous stochastic process. It can be defined as an infinite collection of random variables, with the particularity that any finite subset of these random variables have a joint Gaussian distribution. Being a stochastic process, a GP defines a distribution over functions. A GP is fully characterised by a mean function $m(\cdot)$ and a covariance function (also called kernel) $k(\cdot, \cdot)$, such that any real process $f(\cdot)$ drawn from this GP is described by:

$$
\begin{aligned}
f(t) \quad &\sim GP(m(t), k(t, \cdot)) \\
\text{with} \quad m(t) \quad &= \mathbb{E}[f(t)] \\
\text{and} \quad k(t, t') \quad &= \mathbb{E}[(f(t) - m(t))(f(t') - m(t'))] \;.
\end{aligned}
\tag{2}
$$

A common choice of kernel is the squared-exponential kernel, also called Radial Basis Function, or RBF kernel:

$$
k(t_i, t_j) = \sigma^2 \cdot \exp\left(-\frac{||t_i - t_j||^2}{2\lambda^2}\right) \;.
\tag{3}
$$

This kernel generates smooth (infinitely differentiable) sample paths. The kernel has two hyper-parameters: the output-scale $\sigma^2$ and the lengthscale $\lambda$. $\sigma$ describes how far from the GP's mean will $f$ typically vary. The lengthscale $\lambda$ describes how fast the correlation decreases with the distance between two points. Short lengthscales describe rapidly varying functions, while longer lengthscales describe slowly varying functions. We refer the reader to Williams and Rasmussen [23] for a more detailed introduction to GPs.

GP regression is a Bayesian approach. It consists in defining a GP prior over the modeled continuous process $f$, whose distribution is then updated to a posterior distribution in regard to the available data. A general GP regression model reads:

$$
\forall i \in [\![1, N]\!] \begin{cases} y_i = f(t_i) + \epsilon_i \\ f(\cdot) \sim GP(m(\cdot), k(\cdot, \cdot)) \\ \epsilon_i \text{ an error term with some distribution} \end{cases}
\tag{4}
$$

This general model can be refined by defining $m(\cdot)$, $k(\cdot, \cdot)$ and the distribution of the error term $\epsilon$.

NPARC [6], the current state-of-the-art method to analyse TPP-TR datasets, can be rewritten in the form of a GP regression to simplify its comparison with the presented model. In NPARC, $m$ is chosen to be parameterized by a sigmoid constrained between 0 and 1:

$$
m(t) \equiv S_{a,b,p}(t) = \frac{1 - p}{1 + \exp\left(b - \frac{a}{t}\right)} + p \;.
\tag{5}
$$

The fact that $m$ is constrained between 0 and 1 is linked to the use of Fold Changes (FCs) as scaled observations, i.e for all $r, c, p$, the scaling factor $\rho_{pcr}$ from Eq (1) is chosen to be the raw abundance at the lowest temperature $\gamma_1^{pcr}$. The sigmoidal assumption comes from the chemical denaturation theory [16] which predicts a sigmoidal shape of the melting curve under simplifying assumptions. The kernel $k$ could be seen as a RBF kernel (Eq (3)) with a null output-scale, i.e. $k(t, t') \equiv 0 \; \forall t, t'$. Finally, the error term is chosen to be independent and identically

normally distributed with variance $\beta^2$. In the original paper, parameters $\{a, b, p, \beta\}$ are optimised via nonlinear least square estimation [24] which is equivalent to maximum likelihood estimation.

Two caveats of NPARC addressed by Fang et al. [17] are the lack of uncertainty quantification for the model parameters estimation, and the limited validity of the null distribution approximation. A Bayesian version of the NPARC model is proposed that offers uncertainty quantification. Moreover, statistical assessment of changes in melting behaviour is performed by comparing the posterior model probabilities (null vs alternative model) given the data.

However, the main limitation of NPARC remains the sigmoidal assumption. Indeed, as mentioned in several recent works [17–19], the use of NPARC limits the investigation of non-sigmoidal melting curves, which are likely to carry important biological information. This is a substantial limitation since it has been estimated [17] that up to 20% of the protein-level TPP datasets can show unconventional behaviours. Moreover, our exploration of the published phospho-peptide-level TPP-TR dataset [11] presented here suggests that about 44% of the phospho-peptides studied show non-sigmoidal behaviour (see Fig 5A). To deal with this limitation, Fang et al. [17] proposed an updated version of their Bayesian sigmoid model, called the Bayesian semi-parametric model, in which an additional term, a zero-centered GP, captures correlated residuals and model deviations from the sigmoidal assumption. This model can be fully described as a GP regression, with $m$ defined as in Eq (5), k being a standard RBF kernel (Eq (3)) and $\forall i \quad \epsilon_i \overset{iid}{\sim} \mathcal{N}(0, \beta^2)$. However, the proposed Bayesian inference and model selection remain challenging and complex, requiring marginal likelihood approximation and the choice of a non-negligible number of priors. We refer the reader to Appendix A in S1 File for a more detailed discussion.

Fang et al. [17] allowed for non-sigmoidal curves by adding a GP to the sigmoidal mean function. We relax the sigmoidal assumption by setting the mean to zero ($m \equiv 0$) and instead use correlations across replicates and conditions to capture typical behaviours through hierarchical modelling. This simplification over the Bayesian semi-parametric model allows to infer the GP parameters via type II maximum likelihood estimation (type II MLE), hence making the model inference and selection more straightforward and computationally efficient. While the error term remains independent and identically normally distributed with variance $\beta^2$ as in the previous models, the definition of the kernel $k$ is obtained by combination of multiple kernels, corresponding to the different levels of a hierarchical model. This hierarchy, introduced gradually in the next sections, offers a great flexibility to the model, allowing to deal with multiple conditions and more complex TPP-TR protocols.

## Hierarchical Gaussian process (HGP) models

In this section, we describe the hierarchical Gaussian process (HGP) model adapted from Hensman et al. [25] and originally applied to gene expression time-series. Considering a TPP-TR experiment on the protein level with two conditions, namely a control and a treatment condition, we propose to build step-by-step the different layers of this hierarchy.

For the ease of notation, we consider in the following a unique protein $p$, with the same number $R$ of replicates in both conditions, and the same number of observations $N$ for each replicate. In practice, the model is more general, and can be applied to any number of replicates per condition and different number of observations per replicate.

**Modelling replicates.** Naturally, replicates of a specific condition are expected to be strongly correlated to each other. Indeed, being independent measurements of a unique process, namely the melting behaviour of the protein in this condition, we expect them to share a similar trend, up to some biological variation and technical noise. We are interested in

modeling this trend, regardless of its shape. To this aim, we propose to model this trend by a function denoted $g_c$, on which we assume a zero-centered GP prior with a RBF kernel. Subsequently, we proceed similarly as done by Hensman et al. [25]. To model the deviations of individual replicates from the trend $g_c$, we model the melting behaviour of each replicate $r$ of the condition $c$ by a function $f_{cr}$ distributed according to a GP centered in $g_c$ with a RBF kernel. Furthermore, considering that some replicates could be more variable than others, we suggest to introduce a replicate-wise output-scale $\sigma_{f_{cr}}$ parameter. The model, at this stage, reads:

$$
\begin{aligned}
\forall\ i \in [\![1, N]\!] \\
\forall\ r \in [\![1, R]\!] \\
\forall\ c \in [\![1, C]\!]
\end{aligned}
\begin{cases}
g_c \sim GP(0, k_g(t, \cdot | \lambda_1)) & \textit{conditions} \\
f_{cr} \sim GP(g_c, k_{f_{cr}}(t, \cdot | \lambda_2)) & \textit{replicates} \\
y_{cri} = f_{cr}(t_i) + \epsilon_{cri} & \textit{observations} \\
\epsilon_{cri} \overset{iid}{\sim} \mathcal{N}(0, \beta^2)
\end{cases}
\tag{6}
$$

Given that all the kernels we use in our models are RBF kernels, we propose the following notations, applicable to the whole text:

$$
\begin{aligned}
k_\kappa(t, t' | \lambda) &= \sigma_\kappa^2 \cdot k(t, t' | \lambda) \\
\text{with}\quad k(t, t' | \lambda) &= \exp\left(-\frac{||t - t'||^2}{2\lambda^2}\right).
\end{aligned}
\tag{7}
$$

With this notation, the lengthscale and output-scale parts of the kernel are split. This will be convenient to specify models where lengthscale but not output-scale parameters are shared between levels of the hierarchy.

**Modelling conditions.** Information between replicates of the same condition should be shared, as biological replicates are expected to reflect the typical melting behaviour of the protein in this condition. Similarly, information between replicates of different conditions may also be usefully shared. We first consider the situation where the treatment wouldn't affect the thermal stability of the protein. In this case, similar melting curves in both control and treatment conditions are expected. Hence, replicates from any condition could be seen as independent observations of the same underlying melting behaviour. Sharing information across conditions would thus be beneficial for the model fitting. In contrast, if the treatment affects the thermal stability of the protein, the melting behaviours observed in the different conditions are expected to be less similar. This level of similarity can be understood as an indicator of the amount of perturbation induced on the thermal stability of the protein, and can be of biological interest.

To model these expectations, we propose to add a layer to the hierarchical model described in Eq (6). We denote by $h$ the underlying trend shared between conditions, and assume a zero-centered GP prior on $h$. Consequently, model (6) is updated by centering on $h$ the GP prior over $g_c$. Updating the mean of the GP prior for $g_c$ models our assumption that the melting curves observed in each condition can be seen as deviations from the common melting behaviour $h$. The parameter $\sigma_g$ can thus be understood as a measure of how different is the melting behaviours $g_c$ of condition $c$ from the common trend $h$. Hence, this kernel allows to indirectly capture the level of similarity between the functions $g_c$ of the different conditions. The complete hierarchical GP model reads:

_Three-level hierarchical model for protein-level TPP-TR datasets analysis_:

$$\begin{aligned}
\forall\; i \in [\![1, N]\!] \\
\forall\; r \in [\![1, R]\!] \\
\forall\; c \in [\![1, C]\!]
\end{aligned}
\begin{cases}
h \sim GP(0, k_h(t, \cdot | \lambda_1)) & \textit{protein} \\
g_c \sim GP(h, k_g(t, \cdot | \lambda_1)) & \textit{conditions} \\
f_{cr} \sim GP(g_c, k_{f_{cr}}(t, \cdot | \lambda_2)) & \textit{replicates} \\
y_{cri} = f_{cr}(t_i) + \epsilon_{cri} & \textit{observations} \\
\epsilon_{cri} \overset{iid}{\sim} \mathcal{N}(0, \beta^2)
\end{cases} \tag{8}$$

**Link to the multi-task GP regression framework.** Thanks to the linearity of the hierarchical model, the likelihood of model (8) for protein $p$ can be rewritten as [25]:

$$Y_p | T_p, \theta_p \sim \mathcal{N}(\mathbf{0}, \Sigma_p + \beta_p^2 I_{N_p}) \tag{9}$$

where $\mathbf{0} \in \mathcal{R}^{N_p}$ is the null vector, $I_{N_p} \in \mathcal{R}^{N_p \times N_p}$ the identity matrix, $\Sigma_p \in \mathcal{R}^{N_p \times N_p}$, and $Y_p, T_p, N_p$ and $\theta_p$ as described in Table 1. We show (see detailed derivations in Appendix B in S1 File) that $\Sigma_p$ can be interpreted as a special matrix product of an index kernel $K^y$ and a correlation matrix $K^{t,\lambda}$, typically corresponding to the semantics and models introduced for multi-task GP regression [26]. We argue that breaking down the complexity of the covariance matrix $\Sigma_p$ in terms of index kernels greatly simplifies the understanding and interpretation of the model. Hence, extension of the model to deeper hierarchies or different experimental setups is considerably facilitated. We propose an illustration of the model and its associated covariance matrix $\Sigma_p$ in Fig 2.

## Hypothesis testing framework

A key aspect of the TPP-TR experiments is to determine the set of differentially melting proteins, assumed to be the set of proteins directly or indirectly affected by the treatment(s) (among which one can find: addition of drugs, metabolites, nucleic acids, genetic perturbations, etc.). For simplicity, we start by discussing the case of a TPP-TR experiment involving only two conditions, namely a control and a treatment, with $c \in \{c_1, c_2\}$. The multiple conditions setup is discussed in a subsequent section.

The approach presented in this work lies at the boundary between Bayesian and frequentist statistics. By nature, a GP regression is a Bayesian approach due to the use of GP priors. However, we proceed to a frequentist inference of the model's hyper-parameters via type II MLE, and further develop a hypothesis testing framework, presented hereafter. We begin by introducing the concept of the null hypothesis. Our statistic $\Lambda$ is then presented, and we explain why this statistic is appropriate to provide evidence of differential melting behaviours. The distribution of this non-standard statistic having no known analytical solution, we therefore propose to approximate this null distribution via a sampling method.

**Null hypothesis.** Similarly to NPARC [6] and the Bayesian semi-parametric model [17], we introduce the concept of hypothesis testing by defining two hypotheses:

- Under the _null hypothesis_, we assume that the melting behaviour of protein $p$ follows the same dynamic in both conditions. This is equivalent to assume that

$$g_{c_1} = g_{c_2} \equiv g_{c_0} \; . \tag{10}$$

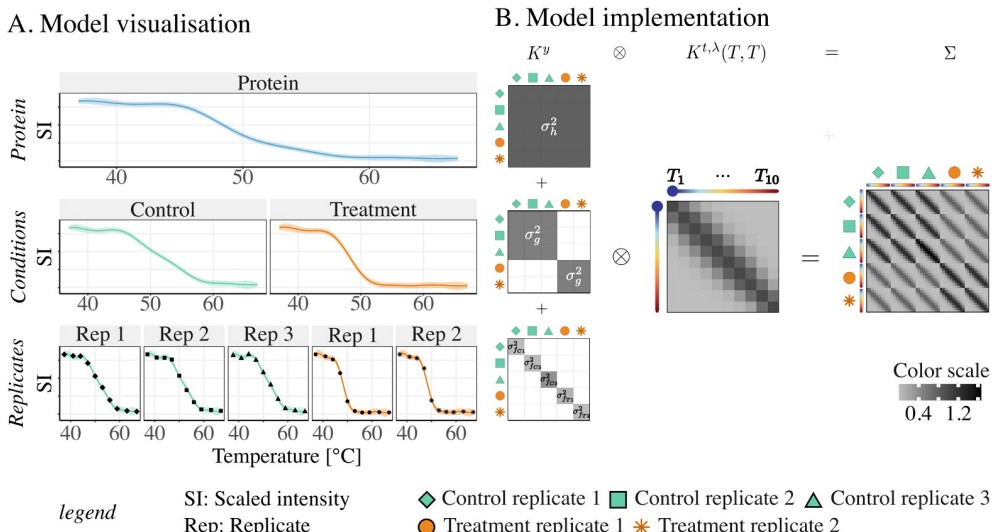

**Fig 2. Implementation of the hierarchical GP model using the multi-task GP regression framework.** The three-level hierarchical model (Eq (8)) is illustrated on a hypothetical protein presenting different numbers of replicates in the control and treatment conditions, under the simplifying assumption of synchronous observations for all replicates of all conditions. (A) A visualization of the model. Melting curves are fitted to observations of each replicate (bottom level). The condition-wise melting curves (second level) captures the underlying melting behaviours common to replicates of a condition. These condition-wise melting curves can be seen as deviations from the protein-wise melting curve depicted on the top of the hierarchy. (B) Schematic visualisation of the resulting covariance matrix $\Sigma$, expressed as a special matrix product between $K^y$, the sum of the index kernels, and the correlation matrix $K^{t,\lambda}(T, T)$, evaluated at the set of temperatures $T = (T_1, \ldots, T_{10})$. This decomposition of the matrix links the hierarchical GP model to the multi-task GP regression framework. Under the simplifying assumption of synchronous observations, the matrix product is a kronecker product. This product is easier to visualize than the Hadamard product obtained in case of asynchronous observations (see Appendix B in S1 File for details).

Using this assumption, model (8) becomes the so-called *joint* model $\mathcal{M}_0$:
*Joint model for protein-level TPP-TR datasets analysis*:

$$
\begin{aligned}
\forall\ i \in [\![1, N]\!] \\
\forall\ r \in [\![1, R]\!] \\
\forall\ c \in [\![c_1, c_2]\!]
\end{aligned}
\left\{
\begin{array}{ll}
h \sim GP(0, k_h(t, \cdot|\lambda_1)) & \text{protein} \\
g_{c_0} \sim GP(h, k_g(t, \cdot|\lambda_1)) & \text{conditions} \\
f_{cr} \sim GP(g_{c_0}, k_{f_{cr}}(t, \cdot|\lambda_2)) & \text{replicates} \\
y_{cri} = f_{cr}(t_i) + \epsilon_{cri} & \text{observations} \\
\epsilon_{cri} \overset{iid}{\sim} \mathcal{N}(0, \beta^2)
\end{array}
\right.
\tag{11}
$$

- Under the *alternative hypothesis*, we assume that the melting behaviour of the protein in each condition might be different, and the model $\mathcal{M}_1$, referred to hereafter as *full* model, is given by model (8).

**The choice of the statistic $\Lambda$.** In a Bayesian framework, a key quantity for model selection is the Bayes factor [27], which evaluates the evidence in favor of the null hypothesis (see Appendix A in S1 File). A frequentist counterpart of the Bayes factor is the Likelihood ratio test statistic, defined as the ratio of the data likelihood under the two compared models, given that the model parameters have been estimated by MLE. In GPMelt, we propose to use a statistic related to, *but different from*, the Likelihood ratio test statistic. This statistic, denoted $\Lambda$, is

defined as the ratio of the log marginal likelihood of our joint model $\mathcal{M}_0$ and full model $\mathcal{M}_1$:

$$\Lambda_p = -2 \cdot \log\left(\frac{p(Y_p|T_p, \theta_p^{MLE-II}, \mathcal{M}_0)}{p(Y_p|T_p, \theta_p^{MLE-II}, \mathcal{M}_1)}\right) \ , \tag{12}$$

where $\theta_p^{MLE-II}$ are the hyper-parameters estimated via type II MLE for the full model. A key difference between a standard Likelihood ratio test statistic and our statistic is the use of the same set of hyper-parameters for the joint and full models.

Three main reasons justify this choice of statistic. First, the *log marginal likelihood* of the data is a key quantity in GP regression (see Appendix A in S1 File) and is readily accessible. Secondly, this statistic is interpretable, as being the log ratio of the likelihood of the data under the two models we aim to compare. Finally, this statistic is inspired from a similar statistic previously shown to be an appropriate similarity measure for time series modeled by GPs [28] (see Appendix C in S1 File).

Furthermore, this statistic presents multiple advantages in the GPMelt setting, further detailed in Supporting Information B in S1 File. Most importantly, the parameters of the joint model given by Eq (11) wouldn't all be identifiable. Additionally, the computational cost linked to the model fitting is significantly reduced, especially in presence of multiple conditions (see subsequent section about extension of GPMelt to the multiple conditions case), as a *unique* fitting process is required per protein. Finally, the expression of $\Lambda_p$ can be simplified as a ratio between the likelihood of two conditional multivariate normals of dimensions $N_Z$, with $N_Z = N_{Control} + N_{Treatment}$ (see Appendix C in S1 File). This is especially advantageous in presence of a large number of conditions, where $N_Z \ll N_p$.

**Approximation of the null distribution.**   The proposed statistic, which is non-standard, has an unknown distribution under the null hypothesis. Therefore, this unknown distribution, also called null distribution of the statistic, has to be approximated to further determine the statistical significance of large values of $\Lambda$.

To this aim, we suggest to proceed similarly as done in Phillips et al. [29], and to sample dynamics according to the multivariate normal describing the observations distribution under the joint model (Eq (11)). More details about this sampling method are given in Supporting Information B, Table E and Fig B in S1 File.

## Model extensions

In this section, we illustrate the versatility of the introduced model. First of all, the model can be applied to multiple conditions by *horizontally* expanding the hierarchical model. Secondly, *vertical* expansion of the hierarchy allows to tackle more complex TPP-TR setups and biological questions.

**Multiple conditions.**   We extend the hypothesis testing framework in presence of $C \geq 2$ conditions. Considering a protein-level TPP-TR datasets with three conditions, and defining without loss of generality $c_1$, $c_2$ the two treatment conditions and $c_3$, also denoted by *Ctrl*, the control condition, we aim to test if $c_j$ has a different melting behaviour than *Ctrl*, for $j \in [\![1, 2]\!]$. Fig 3 illustrates this scenario. Generalization of the described procedure to more conditions naturally follows.

Similarly to the previously introduced joint model (Eq (11)), comparing conditions *Ctrl* and $c_j$ requires to define a null hypothesis, in which the melting behaviours of *Ctrl* and $c_j$ are assumed to follow a common trend, denoted hereafter by $g_{c_0^{Ctrl,j}}$. It can be noticed that this assumption doesn't affect our belief about $c_{j'}$ dynamics (for $j \neq j'$), leading to the following

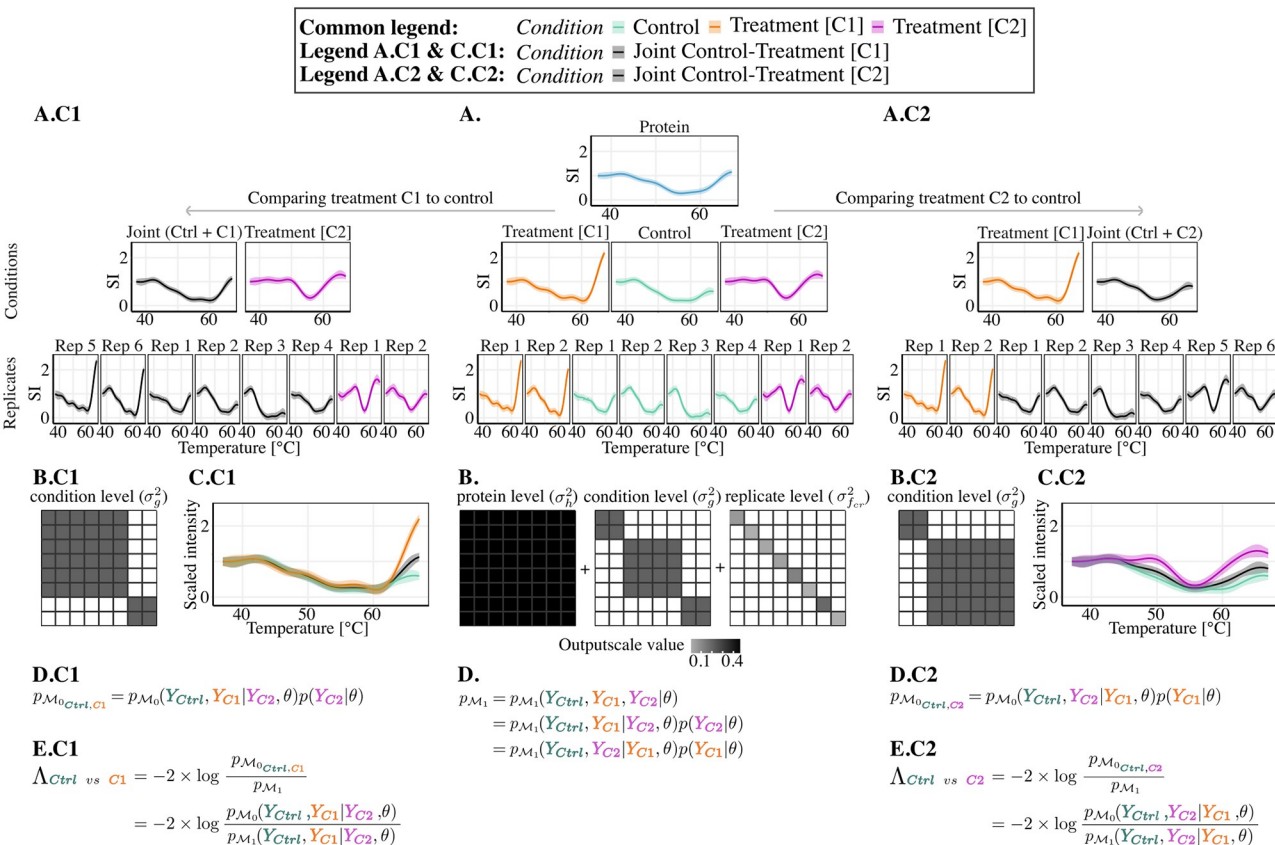

**Fig 3. Principle of GPMelt in presence of multiple conditions.** (SI: Scaled Intensity) Fitting the full model (A, B) is enough to access all the information required to test all possible null hypotheses. The illustration is based on protein SFRS9 of the Dasatinib dataset [1]. The aim of this experiment is to determine changes in melting behaviours upon dasatinib treatment, a BCR-ABL inhibitor. In the experimental set up, the control condition (no treatment) is compared to two treatment concentrations, $0.5\mu M$ and $5\mu M$. For clarity in the figure, treatment concentration of $0.5\mu M$ is referred to as condition "C1" and treatment concentration of $5\mu M$ is referred to as "C2". Control condition is abbreviated by "Ctrl". **(A,B and D): Full model** $\mathcal{M}_1$. (A) Hierarchical model corresponding to Eq (8), in which each condition (middle row) is assumed to present a distinct melting behaviour, this behaviour being a deviation from the main protein-wise melting behaviour (top row, blue curve). The fitting of the observations (last row) under this model provides estimated values for output-scales $\sigma_h$, $\sigma_g$ and $\sigma_{f_{cr}}$. Similarly as in Fig 2, the estimated output-scales can be represented using the index kernels of the multi-task regression framework, as depicted in panel (B). Under this model, the likelihood of the observations is given by $p_{\mathcal{M}_1}$ (D) and further detailed in Appendix C in S1 File. **(A.C1 to E.C1): Comparing treatment C1 with control**. We aim to compare the melting behaviour of this protein in the control condition (green curve) vs the treatment condition C1 (orange curve). A visualisation of this comparison is provided in panel (C.C1). Under the proposed testing framework, the joint model $\mathcal{M}_{0_{Ctrl,C1}}$ assumes that treatment C1 and control conditions have the same melting dynamic, and group them into a "joint" condition (grey curves in (A.C1, C.C1)). (B.C1) Mathematically, this joint model is obtained by changing the structure of the index kernel corresponding to the condition level. More precisely, the "joint" condition, grouping "C1" and "Ctrl", is represented by the upper block in the matrix. Importantly, the values of the output-scales $\sigma_h$, $\sigma_g$ and $\sigma_{f_{cr}}$ remain unchanged: there is no need to *re-estimate* the parameters of this model. Moreover, the modelling of condition "C2" is not affected by this joint model, as can be seen in (A.C1) and (B.C1). (D.C1) The likelihood of the observations under this model is given by $p_{\mathcal{M}_{0_{Ctrl,C1}}}$. (E.C1) The statistic $\Lambda$ used to statistically assess the significance of melting behaviour changes is given by $\Lambda_{Ctrl\ vs\ C1}$. **(A.C2 to E.C2): Comparing treatment C2 with control**. Similarly, we illustrate the procedure to compare the protein's melting behaviours between treatment C2 (pink curve) and control (green curve) conditions. Under this model, conditions "C2" and "Ctrl" are grouped together in the "joint" condition, while condition "C1" is unaffected (A.C2, B.C2). Melting behaviours changes are depicted in panel (C.C2). The likelihood of the observations under this model, $p_{\mathcal{M}_{0_{Ctrl,C2}}}$ (D.C2), and the associated statistic $\Lambda_{Ctrl\ vs\ C2}$ (E.C2) are given.

joint model $\mathcal{M}_{0_{Ctrl,j}}$:

$$
\forall\ i \in [\![1,N]\!] \\
\forall\ r \in [\![1,R]\!]
\begin{cases}
h \sim GP(0, k_h(t, \cdot | \lambda_1)) & \textit{protein} \\[2mm]
g_{c_0^{Ctrl,j}} \sim GP(h, k_g(t, \cdot | \lambda_1)) & \textit{cond. Ctrl \& } c_j \\[2mm]
g_{c_{j'}} \sim GP(h, k_g(t, \cdot | \lambda_1)) & \textit{cond. } c_{j'} \\[2mm]
f_{cr} \sim
\begin{cases}
GP(g_{c_0^{Ctrl,j}}, k_{f_{cr}}(t, \cdot | \lambda_2)) \textit{if } c \in \{Ctrl, c_j\} \\[2mm]
GP(g_{c_{j'}}, k_{f_{cr}}(t, \cdot | \lambda_2)) \ \textit{if } c = c_{j'}
\end{cases} & \textit{replicates} \\[4mm]
y_{cri} = f_{cr}(t_i) + \epsilon_{cri} & \textit{observations} \\[2mm]
\epsilon_{cri} \overset{iid}{\sim} \mathcal{N}(0, \beta^2)
\end{cases}
\tag{13}
$$

The statistic $\Lambda$ for the comparison *Ctrl* vs $c_j$ is given by (see Appendix C in S1 File for detailed derivations):

$$
\Lambda_{Ctrl,c_j} = -2 \cdot \log \frac{p(Y_{Ctrl}, Y_{c_j} | Y_{c_{j'}}, T_{Ctrl}, T_{c_j}, T_{c_{j'}}, \theta^{MLE-II}, \mathcal{M}_{0_{Ctrl,j}})}{p(Y_{Ctrl}, Y_{c_j} | Y_{c_{j'}}, T_{Ctrl}, T_{c_j}, T_{c_{j'}}, \theta^{MLE-II}, \mathcal{M}_1)} \ .
\tag{14}
$$

where $\theta^{MLE-II}$ are the hyper-parameters estimated via type II MLE for the full model $\mathcal{M}_1$ (Eq (8)), and the following simplified notations are used: $Y_c \equiv Y_{pc}$, $T_c \equiv T_{pc}$ for $c \in \{Ctrl, c_j, c_{j}'\}$. Appendix C in S1 File shows that the expression of $\Lambda$ can always be expressed as the ratio of two conditional normal distributions. These conditional distributions describe the probabilities of the compared conditions (here *Ctrl* and $c_j$) given all non-compared conditions (here only $c_j'$).

**Deeper hierarchy.** The expansion of the model to deeper hierarchies is motivated by the analysis of peptide-level TPP-TR datasets. However, it could be applied to any TPP-TR experimental setups where it is appropriate to introduce an additional layer to better capture existing similarities between observations.

Unlike protein-level melting curves, averaged over the measurements of all tryptic peptides corresponding to a single protein entry in the protein database, peptide-level melting curves exhibit greater replicate to replicate variability, higher levels of noise and more frequent unconventional melting behaviours. Moreover, peptides associated to a single protein entry might also present various melting behaviours. Indeed, proteins in cells typically present multiple sub-populations of proteoforms [30]. These proteoforms correspond to possible molecular modifications affecting the protein product of a single gene, among which are found genetic variations, alternative splicing and post-translational modifications (PTMs) [15]. Different proteoforms can have different cellular localisations and/or functions, which will be reflected in differences in melting behaviours [14]. Thus, measured peptides mapped to a single protein entry can originate from different proteoforms. To account for this higher level of data complexity, we introduce a new level in our hierarchical model presented in Eq (8). Furthermore, the meaning of *condition* is extended to this peptide-level measurements setting. When analyzing peptide-level TPP-TR dataset, the goal is generally to determine if a given group of peptides present a similar melting behaviour than another group. As an example, the melting behaviour of one proteoform could be compared to another. Hence, a group of peptides can be seen as playing a similar role than a *condition* in the previously introduced models.

Given a protein $p$, we consider the task of modelling the individual measurements of $\Pi$ peptides denoted by $\pi_j$, with $j \in [\![1, \Pi]\!]$. Each peptide $\pi_j$ presents a given number of replicates $R_{\pi_j}$. We further assume that these peptides can be grouped into $C$ conditions (e.g. *proteoforms*), that we aim to compare. In the following and for notational convenience, all peptides are assumed to have the same number of replicates $R$ measured at the same $T$ temperatures. This doesn't have to be the case in practice.

Following the same logic used to construct the three-level hierarchical model (Eq (8)), we first model the melting curves $\eta_{c\pi jr}$ of replicates of peptide $\pi_j$, by a GP prior centered in a peptide-specific function $f_{c\pi_j}$. This peptide $\pi_j$ is further believed to be part of condition $c$, and thus $f_{c\pi_j}$ is modeled by a GP prior centered on a condition-specific function denoted $g_c$. All condition-specific functions are modeled by a GP prior centered on a function $h$, itself modeled by a zero-centered GP. The full model reads:

$$
\forall\, i \in [\![1, N]\!] \\
\forall\, r \in [\![1, R]\!] \\
\forall\, j \in [\![1, \textstyle\prod]\!] \\
\forall\, c \in [\![1, C]\!]
\begin{cases}
h \sim GP(0, k_h(t, \cdot | \lambda_1)) & protein \\
g_c \sim GP(h, k_g(t, \cdot | \lambda_1)) & conditions \\
f_{c\pi_j} \sim GP(g_c, k_{f_{c\pi_j}}(t, \cdot | \lambda_2)) & peptides \\
\eta_{c\pi_j r} \sim GP(f_{c\pi_j}, k_{\eta_{c\pi_j r}}(t, \cdot | \lambda_3)) & replicates \\
y_{c\pi_j ri} = \eta_{c\pi_j r}(t_i) + \epsilon_{c\pi_j ri} & observations \\
\epsilon_{c\pi_j ri} \overset{iid}{\sim} \mathcal{N}(0, \beta^2)
\end{cases}
\tag{15}
$$

In this model, a third lengthscale $\lambda_3$ has been introduced, along with replicate- and peptide-specific output-scales $\sigma_{\eta_{c\pi_j r}}$ and $\sigma_{f_{c\pi_j}}$. A discussion about model complexity and constraints is provided in Supporting Information A in S1 File. Fig Md in S1 File illustrates a constrained version of this model.

## Implementation

GPMelt's framework implementation is based on the Python [31] package GPyTorch [32], accompanied by a Nextflow [33] pipeline. This setup ensures simple installation, supports parallel computation, and offers a portable solution for deployment on both local computers and high-performance computing (HPC) clusters. An overview of the resource requirement of the implementation used to generate the results is illustrated in Fig W in S1 File for the ATP 2019 and phosphoTPP datasets.

## Results

Hereafter, the number of levels of the hierarchical model used within the GPMelt statistical framework (three-level: Eq (8), four-level: Eq (15)) is specified for each analysis.

We start by showing the validity of the GPMelt framework using a simulation study. Subsequently, to benchmark the GPMelt framework with a three-level HGP model on real data, we reanalysed four published protein-level TPP-TR datasets (called thereafter ATP 2015 [34], Staurosporine 2014 [1], Panobinostat [8] and Dasatinib [1]) on which NPARC [6], the Bayesian sigmoid [17] and the Bayesian semi-parametric [17] models have been previously applied in the corresponding publications. Additionally, we re-analysed protein-level data from two more recent publications on which GPMelt with a three-level HGP model is compared to

**Table 2. Description of the benchmarking datasets.**

| | Dataset | Comparison | Approximation of ground truth | Number of IDs | Other applied methods and references to results (if applicable) |
|---|---|---|---|---|---|
| protein-level TPP-TR two conditions | Staurosporine 2014 [1] | Vehicle vs Staurosporine [20$\mu$M] | GO term: protein kinase activity | 4505 proteins | $T_m$ and NPARC [6] Bayesian sigmoid [17] Bayesian semi-parametric models [17] |
| | Staurosporine 2021 [35] | Vehicle vs Staurosporine [20$\mu$M] | GO term: protein kinase activity | 4403 proteins | $T_m$ [35]; NPARC |
| | ATP 2019 [19] | Vehicle vs Mg-ATP [10 mM] | GO term: ATP-binding proteins | 4772 proteins | NPARC |
| peptide-level TPP-TR multiple conditions | phospho-TPP [11] | Phospho-peptides vs median of non-phosphorylated peptides | Functional scores [36] | 13990 phospho-peptides from 1949 proteins | $T_m$ [11] |
| | phospho-TPP [11] | Comparison of phosphorylation patterns | | 4073 phospho-peptides from 310 proteins | |

NPARC: the ATP 2019 [19] and the Staurosporine 2021 datasets [35]. We focus hereafter on the results concerning the Staurosporine 2014, Staurosporine 2021 and the ATP 2019 datasets. Further results are discussed in S1 File. Furthermore, to exemplify GPMelt with a three-level HGP model on peptide-level TPP-TR datasets, the published phospho-TPP dataset [11] is re-analyzed. Finally, we demonstrate how GPMelt with a four-level HGP model can be used to analyse the phospho-TPP dataset using a different approach. Tables 2 and C in S1 File detail information about the datasets, Table D in S1 File presents the models specifications.

## Simulation study

To validate the newly introduced statistic $\Lambda$, we proceeded to a simulation study, in which we generated synthetic data according to GPMelt full and joint models, for hierarchical models with three or four levels, and with two or three conditions. Additionally, we investigated the effect of the proportion of true differential melting curves in the synthetic dataset, varying this number from 1% to 30%. Finally, we examined the impact of noisy data on GPMelt results, and increased the amount of correlated noise per replicate from 1 to 1000 times more than typically observed in clean real data. Example of samples forming the different synthetic datasets are depicted in Figs C-F in S1 File.

We start by demonstrating the accuracy of GPMelt to detect true differential melting curves using ROC curves (Fig G in S1 File). Subsequently, we show that the FDR is well controlled for datasets up to 100 times more noisy than real clean data (Fig H in S1 File). Finally, we illustrate that the use of $\Lambda$ and the sampling approach to approximate its null distribution provides well-calibrated p-values at normal noise level, calibration which remains fairly robust to an increase of noise (see Figs I-L in S1 File). Supporting Information C in S1 File. provides additional descriptions regarding the synthetic dataset generation and the obtained results.

## Protein-level TPP-TR with two conditions

We start the real datasets analyses by applying GPMelt with a three-level HGP model on the Staurosporine 2014 [1] and ATP 2019 [19] datasets. The results obtained for the Staurosporine 2014 dataset are compared to three of the former methods (NPARC, Bayesian sigmoid and Bayesian semi-parametric models) for which results on this exact same dataset have been published and are readily available online [6, 17]. To benchmark our analysis on the ATP 2019 dataset, we applied NPARC analysis (R package version 1.12). To proceed to the methods

comparison, an *approximate* receiver operator characteristic (ROC) curves is built for each method. The term *approximation* refers to the exact ground truth being unknown, but approximated by the set of proteins expected to be targeted by the treatment. However, proteins in this set are unlikely to all show a significant change in melting behaviour. Furthermore, some proteins which don't belong to this set (either due to incorrect annotation or because they are unexpected (off-)targets of the investigated compounds) might as well present important variations in their melting curves. Taking this into consideration, the set of expected target proteins can be defined using the Gene Ontology (GO) Consortium annotations curated in Uniprot [37]. For Staurosporine 2014, 176 proteins ($N = 4505$) present a kinase activity (annotations downloaded in march 2023). Regarding the ATP 2019 dataset, 573 out of $N = 4772$ proteins were annotated as ATP binding proteins (annotations provided as Supplementary Table S5 from Sridharan et al [19]). The ROC curves are depicted in Fig 4A, with points corresponding to the sensitivity and specificity of NPARC and GPMelt at an $\alpha$-threshold of $\alpha \in \{0.001, 0.005, 0.01, 0.05\}$ on the BH adjusted p-values, resp. a threshold of $1 - \alpha$ on the posterior probabilities of the alternative model for the Bayesian sigmoid and Bayesian semi-parametric models.

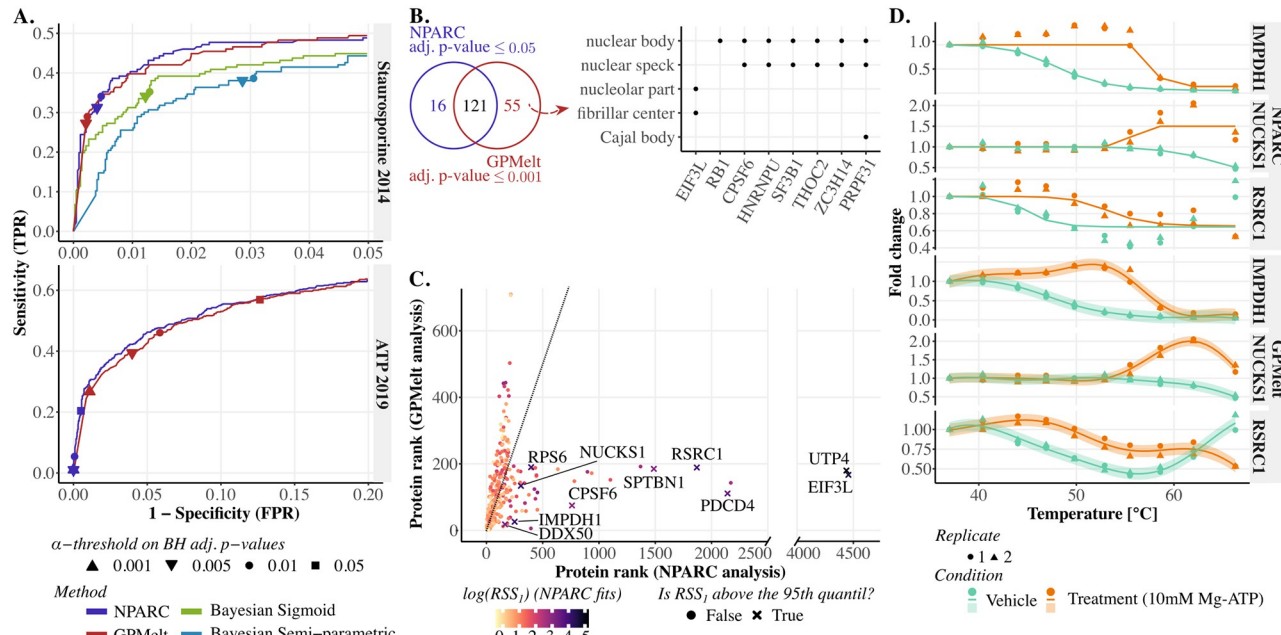

**Fig 4. Including non-sigmoidal melting curves with GPMelt improves the quality of the discoveries for protein-level TPP-TR datasets.** (A) Approximate receiver operator characteristic (ROC) curves comparing the results of NPARC [6], the Bayesian sigmoid and Bayesian semi-parametric models [17] and GPMelt with a three-level HGP model on the Staurosporine 2014 [1] and ATP 2019 [19] datasets. The set of proteins expected to be targeted by the treatments are defined using the Gene Ontology (GO) Consortium annotations curated in Uniprot [37]. For the Staurosporine 2014 dataset, 176 out of 4505 proteins present a kinase activity (annotations downloaded in march 2023). 573 out of 4772 proteins are annotated as ATP binding proteins (using annotations provided as supplementary data in [19]). The points on the curves correspond to the sensitivity and specificity of NPARC and GPMelt at an $\alpha$-threshold of $\alpha \in \{0.001, 0.005, 0.01, 0.05\}$ on the BH adjusted p-values, resp. a threshold of $1 - \alpha$ on the posterior probabilities of the alternative model for the Bayesian sigmoid and Bayesian semi-parametric models. Panels B to D discuss results on the ATP 2019 dataset. (B,left) Overlap of the hits obtained with an $\alpha$-threshold of 0.05 on the adjusted p-values of NPARC and an $\alpha$-threshold of 0.001 on the adjusted p-values of GPMelt. (B,right) Among the 55 hits uniquely selected by GPMelt, eight of them are annotated to be part of membrane-less organelles. The GO cellular compartment terms are provided as supplementary data from [19]. The enrichment analysis is performed with the R package clusterProfiler [38] (v4.8.3), with background defined by the set of proteins identified in the experiment. (C) Comparison of proteins ranking considering NPARC (x-axis) vs GPMelt (y-axis) analysis of the ATP 2019 dataset (for the top 200 proteins of each method). Points are colored according to the Residual Sum of Square of NPARC fits for the alternative model, denoted by $RSS_1$. Crosses represent proteins for which $RSS_1$ is above the 95*th*-percentile (computed across proteins). (D) Examples of proteins low-ranked by NPARC due to non-conventional melting behaviours (see panel C). The melting curves of these proteins are miss-fitted by NPARC due to the inherent sigmoidal assumption. Fig P in S1 File presents additional examples.

For both datasets, the ROC curves of GPMelt and NPARC are almost overlapping. This observation means that, without simplifying assumptions on the melting behaviour shapes, GPMelt performs at least as well as NPARC on datasets presenting a small amount of non-sigmoidal melting curves. In addition, while the Bayesian semi-parametric model also uses GPs to relax the sigmoidal assumption, the ROC curve obtained for this method on the Staurosporine 2014 dataset highlights the lack of specificity of this model compared to GPMelt (similarly illustrated in Fig Oa in S1 File). This point is further investigated in a subsequent section about outliers detection.

Having shown that GPMelt performs better than the Bayesian semi-parametric model while being based on a similar principle of non-sigmoidal curves inclusion, we further focus on comparing NPARC to GPMelt. The ROC curves for the ATP 2019 dataset show that GPMelt is a lot more sensitive than NPARC, capturing 176 hits with an $\alpha$-threshold of 0.001 on the adjusted p-values, for 137 captured proteins using an $\alpha$-threshold of 0.05 on the adjusted p-values of NPARC. The overlap in hits is illustrated in Fig 4B. In this specific dataset, the concentration of the Mg-ATP treatment is particularly high ($10\,mM$), inducing large and reproducible changes in melting curves for a considerable number of proteins. However, the estimation of the parameters of the null distribution of the F-statistic of NPARC method relies on the fact that only a small number of proteins present significant changes in melting behaviours. We suggest that breaking this assumption might result in an incorrect calibration of the p-values, explaining the low sensitivity of NPARC compared to GPMelt for this dataset. Thus, GPMelt can be applied to a broader range of datasets, as the null distribution estimation is more robust to the number of true positives.

Considering that the p-values (and thus adjusted p-values) of NPARC might be incorrectly computed for the ATP 2019 dataset, we further compared the proteins ranking. Fig 4C considers the top 200 proteins captured by each method, with each protein being colored according to the Residual Sum of Square of NPARC fits for the alternative model, denoted by $RSS_1$. More precisely, the alternative model of NPARC consists in fitting two independent sigmoids to the data, one sigmoid for the control and one for the treatment condition. $RSS_1$ is the sum of the residuals of these two fits. The larger $RSS_1$, the less likely the sigmoidal assumption is valid for at least one condition (control or treatment) of this protein. Proteins with $RSS_1$ values exceeding the $95th$-percentile (computed across proteins) are depicted by crosses. With this plot, we aim to illustrate that numerous proteins were low-ranked by NPARC due to non-conventional melting behaviours. Fig 4D illustrates three of these proteins. Although the shape of the melting curves are very diverse, they are consistently well fitted by the hierarchical Gaussian process model of GPMelt, while at least a part of the curve is miss-fitted by NPARC due to the inherent sigmoidal assumption. Fig P in S1 File presents additional examples.

We previously argued that GPMelt is more sensitive than NPARC on the ATP 2019 dataset, capturing 55 additional proteins using an $\alpha$-threshold five times lower than NPARC (Fig 4B). We show that these hits are biologically meaningful. Among the many observations of Sridharan et al [19], ATP is found to impact binding properties of proteins to nucleic acids, and to be involved in multiple ways to the process of phase separation. Phase separation is a biological phenomenon by which weak interactions of proteins with nucleic acids lead to the formation of macromolecular condensates, among which membraneless organelles [20, 21]. Figs 4B and Od in S1 File show that the 55 proteins uniquely captured by GPMelt present a clear enrichment in proteins annotated to be part of membraneless organelles or macromolecular condensates. Moreover, the GO molecular function enrichment applied on these 55 proteins (Fig Oc in S1 File) illustrates that numerous captured proteins have an ATP-dependent activity or are interacting with nucleic acids (using the GO cellular compartment and molecular function terms provided as supplementary data from Sridharan et al [19]).

**Non-sigmoidal melting behaviours.** We mentioned above that both the Staurosporine 2014 and the ATP 2019 present a small amount of non-sigmoidal melting curves. We illustrate with the Staurosporine 2021 dataset [35] (TMT 11 plex) how the presence of non-sigmoidal melting behaviours impact the validity of NPARC analysis, and how this is solved by incorporating non-sigmoidal melting curves in the analysis thanks to the hierarchical Gaussian process model of GPMelt.

The application of NPARC to the Staurosporine 2021 dataset leads to 95 proteins with a failing fit (vs only 3 for the Staurosporine 2014 and 16 for the ATP 2019 dataset). Moreover, when examining the obtained p-values histogram (see left panel of Fig Qe in S1 File), a peak on the right of the distribution is clearly noticeable. The application of Benjamini and Hochberg (BH) procedure [39] for multiple testing correction assumes a uniform distribution of p-values under the null assumption and hence might result in an incorrect adjustment of NPARC's $p$-values on this dataset. To better understand the origin of this peak, we took a closer look at the observations leading to p-values being nearly one. In most of the cases, extreme p-values correspond to cases where the sigmoid fit fails for the alternative and/or the null model (see Fig Ob in S1 File). Similarly as above, we thus used the $RSS_1$ values obtained from NPARC fits as a proxy to melting curves sigmoidality. By removing proteins with $RSS_1$ values exceeding the $90th$-percentile (computed across proteins) from NPARC results, a proper p-value histogram could be reached (Fig Qe in S1 File, middle panel). However, this leads to the removal of 629 + 95 = 724 proteins (accounting for failing fits), meaning that about 16.5% of the dataset could not be correctly analysed via NPARC. This result further motivates the use of GPMelt for protein-level TPP-TR datasets, as being an inclusive, sensitive and specific statistical method leading to biologically relevant results.

## Peptide-level TPP-TR with multiple conditions

We now illustrate how GPMelt can be used to deal with more than two conditions using a peptide-level TPP-TR dataset. To this aim, we propose to re-analyse the phospho-TPP dataset [11], which compares the melting behaviour of phosphorylated peptides to the melting behaviour of the non-phosphorylated peptides associated to the same entry in the protein database. A schematic visualisation of the data at hand is provided in Fig R in S1 File. Functionally relevant phosphosites are expected to induce a change in melting behaviour, by affecting e.g. the protein configuration or protein interactions [11]. The phospho-peptides entering the analysis present one to multiple phosphorylation sites. Individual phosphorylation site functionality can be predicted using the machine-learning based score from Ochoa *et al.* [36], ranging from 0 to 1, with larger values indicating more functionally relevant phosphosites.

We first show (Fig 5A) that GPMelt, by integrating non-sigmoidal melting curves in the analysis thanks to the three-level HGP model, allows to almost double (1.78) the number of phospho-peptides under study compared to the published $T_m$ analysis. A detailed comparison of the data entering both analyses are given in Fig Sa in S1 File.

Furthermore, the analysis of this dataset illustrates how a very reduced number of fits per protein (one to 34) allows to capture any possibly significant phospho-peptides of a protein. The phospho-TPP dataset contains 13990 phospho-peptides corresponding to 1949 gene names. Among these gene names, 1449 ($\approx$ 74%) are associated to more than one phospho-peptide, and 1828 proteins (about 94%) have at most 20 phospho-peptides. The largest number of phospho-peptides associated to a gene name is 664. In the following analysis, we suggest to fit up to 20 phospho-peptides of a protein together, thus corresponding to a three-level hierarchical model with 21 conditions (counting the control condition being the median over the

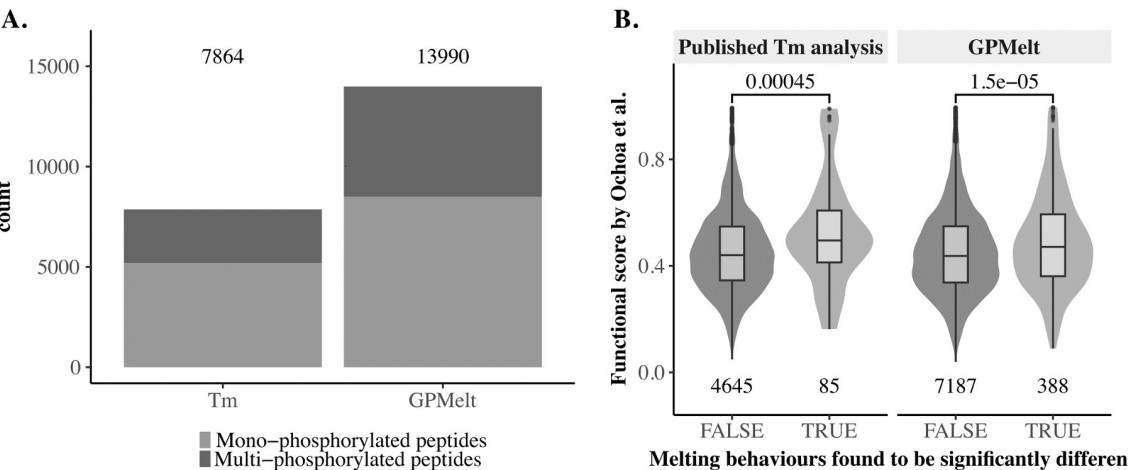

**Fig 5. Including non-sigmoidal melting curves in peptide-level TPP-TR datasets largely increases the number of discoveries.**
Functionally relevant phosphosites are expected to induce a change in melting behaviour by influencing, among others, protein conformations and protein-protein interactions. Mono-phosphorylated peptides functionality can be predicted using the functional score, a machine-learning based score [36], ranging from 0 to 1, with larger values indicating more functionally relevant phosphosites. To detect functionally relevant phosphosites, the melting behaviour of phosphorylated peptides are compared to the melting behaviour of the non-phosphorylated peptides associated to the same entry in the protein database. GPMelt with a three-level HGP model is used to reanalyse the phospho-TPP dataset [11]. (A) Considering non-conventional melting curves in the analysis makes it possible to include almost twice (1.78) as many phospho-peptides compared to the published melting point ($T_m$) analysis. (B) By increasing the phospho-peptides coverage, GPMelt captures about five times more (4.9) mono-phosphorylated peptides than the published $T_m$ analysis, and captures phosphosites associated with significantly higher functional scores than non-captured phosphosites (one-sided Wilcoxon signed-rank test). GPMelt hit selection: any phospho-peptide for which the associated $\Lambda$ value is so extreme that it is strictly above any values belonging to the null distribution approximation ($S = 1e4$ samples per protein). 443 mono-phosphorylated peptides are selected by GPMelt, among which 388 have an associated functional score. The $T_m$ analysis selects 90 mono-phosphorylated peptides, with 85 presenting a known functional score.

non-phosphorylated peptides for this gene name). Details regarding the advantages of this choice can be found in Supporting Information B in S1 File.

Subsequently, we propose to approximate the null distribution of the statistic $\Lambda$ for each protein independently (method D of Table E in S1 File). Indeed, the number of replicates per peptide, as well as the number of peptides per protein, greatly vary between proteins in the phospho-TPP dataset. The value of $\Lambda$ being intrinsically dependent on the number of observations entering the fitting process (see Supporting Information B in S1 File for more details), we thus suggest to proceed independently for each protein. A total number of $S = 1e4$ samples have been obtained for each protein. We define as significant any phospho-peptide for which the estimated value of $\Lambda$ is larger than any values of the corresponding null distribution approximation. The associated BH adjusted p-value for these phospho-peptides is 0.0021. We show in Fig 5B that GPMelt is able to capture five times more mono-phosphorylated peptides than the $T_m$ approach, while capturing phospho-peptides whose associated functional scores are significantly larger than non-captured phospho-peptides (one-sided Wilcoxon signed-rank test). Taking multi-phosphorylated peptides into account, GPMelt captures a total of 648 phospho-peptides, while the $T_m$ approach only captures 129.

The proposed null distribution approximation is very sensitive, at the cost of being computationally expensive. A cheaper approach, corresponding to method E of Table E in S1 File, is illustrated in Supporting Information B and Fig S in S1 File. In this approach, proteins presenting the same number of phospho-peptides are grouped, to compute a *group-wise* null distribution approximation.

### Extension to a four-level hierarchical model

The interpretation of the results of the phospho-peptide approach presented in the previous paragraph present some limitations. Especially, we can consider the case of two phospho-peptides presenting the same phosphorylation patterns, but showing significant differences in their melting behaviours. In this case, a consensus about the functionality of the concerned phosphosites cannot be reached.

The presence of proteoforms in cells provide the main explanation for this phenomena. As mentioned earlier, proteoforms design the set of all proteins originating from a unique gene from a species, and which differ in sequence (e.g due to alternative splicing) and/or in site-specific features (e.g PTMs or single-nucleotide polymorphisms (SNP)) [30]. Multiple sub-populations of proteoforms typically coexist in a cell, and serve different purposes, potentially located in different sub-cellular compartments.

Furthermore, the principle of PTMs cross-talk [40] could also play a role. More precisely, phosphorylation cross-talk investigates phospho-sites cooperation. If phospho-sites act cooperatively, the combination of phospho-sites affects the function of the protein distinctively from the effect of the individual phospho-sites.

Additionally, not all phosphorylation events affect proteins characteristics in the same way. Some work even discuss the idea that only a small fraction of phosphosites could actually be functional [11, 36].

With these information at hand, we consider the TPP protocols, in which proteins are firstly digested in tryptic peptides before being measured via MS. The assignment of measured peptides to proteins is done via database search. It is currently not possible to assign peptides to proteoforms during this identification process. Furthermore, we consider two phospho-peptides $p_1$ and $p_2$. We assume that these peptides span an overlapping sequence of amino acids from the protein, with the sequence localised between amino acids $s_i$ and $e_i$, for $i \in \{1, 2\}$. Without loss of generality, we assume that $s_1 \leq s_2 < e_1 \leq e_2$. Additionally and without loss of generality, we assume that $s_1 < s_2$ with some phosphorylable residues ($S$, $T$ or $Y$) located between $s_1$ and $s_2$. The phosphorylated states of these residues are thus unknown for peptide $p_2$.

The absence of this information can have two major implications. Firstly, if the phosphorylation events affecting residues located between $s_1$ and $s_2$ act cooperatively with at least one of the observed phosphosites located between $s_2$ and $e_1$, we can expect phospho-peptides $p_1$ and $p_2$ to originate from proteins with different functions, localisations and/or interactions. Secondly, if the phosphorylation of residues located between $s_1$ and $s_2$ have stronger effect on the protein than the phosphorylation events located between $s_2$ and $e_1$, this would lead to the same conclusion about phospho-peptides $p_1$ and $p_2$. Hence, it is not possible to conclude on the functionality of the phosphosites located between $s_2$ and $e_1$ by aggregating the observations coming from $p_1$ and $p_2$.

To deal with this situation, we suggest to adapt the analysis of the phospho-TPP dataset [11], and propose an approach focused on the phospho-sites rather than the phospho-peptides. More precisely, we focus on overlapping peptides, similar to $p_1$ and $p_2$, for which all amino acids located between $s_1$ and $s_2$, similarly between $e_1$ and $e_2$, are *non*-phosphorylable residues. This approach is illustrated in Figs M and N in S1 File with a real example. In this example, 17 peptides spanning a region, denoted as "sub-sequence of interest", are observed. This sub-sequence of interest, equivalent to the previously introduced sequence located between $s_2$ and $e_1$, contains exactly four phosphorylable residues, which are all part of the sequences of the 17 measured peptides. These peptides present nine phosphorylation patterns (among which peptides with no phosphorylation at all), with some of these patterns being shared across peptides,

some being uniquely observed. We further propose to extend the three-level hierarchical model introduced in this paper to a four-level hierarchy (Fig Mc in S1 File), as described in Eq (15). In this hierarchy, the nine different phosphorylation patterns can be seen as conditions.

An interesting feature of this model is that the precise definition of the control condition is not required beforehand: it is sufficient to fit the model depicted in Fig Mc in S1 File to have access to the value of $\Lambda$ of any two-by-two comparisons. Especially, this model offers a way to study phosphorylation cross-talk. To conduct this type of analysis, one might have to compare the melting behaviours between any observed combination of phosphorylation events. Hence, there is not one *control* condition, but a multitude of them. As example, we propose in the analysis presented in Fig N in S1 File to consider the two mono-phosphorylated patterns as *control* conditions, and to investigate how the addition of phosphorylation events on top of these initial phosphorylated sites affect the melting behaviour of the protein. This analysis allows to hypothesis that phosphosites $pS145$ and $pT143$ act cooperatively, but that the phosphorylation of $pS147$ in addition to $pS145$ doesn't affect $pS145$ functionality. Further follow-up experiments would be required to validate these hypotheses, as the present analysis cannot exclude the impact of unseen phosphorylation sites outside of the sub-sequence of interest on the melting behaviour of the peptides.

## Additional model features

**Detection and robustness to outliers.**   As mentioned in the results section for protein-level TPP-TR datasets, the ROC curves presented in Figs 4A and Oa in S1 File highlight the lack of specificity of the Bayesian semi-parametric model. In the corresponding paper [17], the authors argued that the false positive rate (FPR) is not well-defined for these datasets. However, a cautious examination of their hits led us to conclude that this method might also suffer from a significant sensitivity to outliers. To show this, we considered the Staurosporine 2014 dataset and selected the proteins for which the BH adjusted p-values from both NPARC and GPMelt were greater than 0.8, while being captured as significant hits by the Bayesian semi-parametric model at an $\alpha$-threshold of 0.01. We further examined the estimated values of the replicate-specific output-scale $\sigma^2_{f_{cr}}$ obtained by fitting a three-level HGP model to these proteins.

Indeed, outlier observations push the replicate-specific melting curves away from the melting behaviour of the corresponding conditions. As $\sigma^2_{f_{cr}}$ is a direct measure of the deviation of replicate $r$ melting curve from the melting curve of condition $c$, large values of $\sigma^2_{f_{cr}}$ suggest the presence of outlier observations. Hence, we compared the replicate-specific output-scale $\sigma^2_{f_{cr}}$ of these proteins to the $95^{th}$ percentile of the associated distribution (computed considering all proteins of the dataset). Fig 6A points out that the selected proteins all present at least one replicate of one condition with an output-scale $\sigma^2_{f_{cr}}$ above the $95^{th}$ percentile. We further illustrate (Fig 6B) how these large values of $\sigma^2_{f_{cr}}$ directly relate to the presence of one to multiple outlier observation(s) in the corresponding replicate(s).

By accounting for singular and independent variations from the condition-wise trend through the replicate-specific output-scale $\sigma^2_{f_{cr}}$, the presented hierarchical model is more robust to outliers values. Moreover, the HGP model parameters provide the user with an easy way to detect outliers observations through the entire dataset.

**Precise estimation of the Area Between the Curves.**   $\Delta T_m = T_m^{treatment} - T_m^{control}$, previously used to measure the positive (stabilisation effect) or negative (destabilisation effect) shift in melting point induced by the treatment, is only valid for sigmoidal curves. We propose a new measure of the discrepancy between the control and treatment curves, denoted by the *Area*

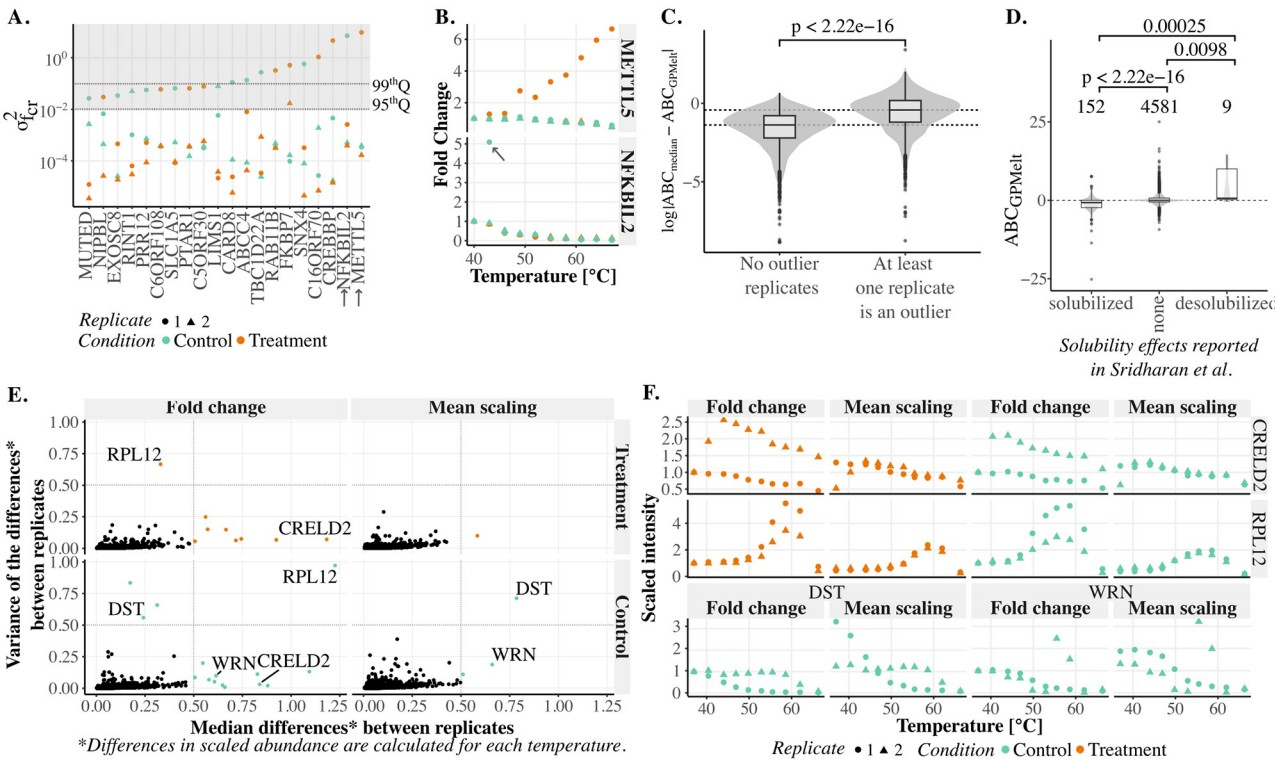

**Fig 6. Additional model features.** Panels A to C present results from the Staurosporine 2014 dataset [1], panels D to F from the ATP 2019 dataset [19]. **(A-B) Detection of outliers** (A) Subset of proteins presenting a BH adjusted p-value superior or equal to 0.8 according to NPARC and GPMelt methods, but an alternative model posterior probability larger than 0.99 according to the Bayesian semi-parametric model. For each of these proteins, the plot represents on a log scale the estimated values of the output-scale parameters $\sigma^2_{f_{cr}}$ obtained from the three-level HGP model, with the shape corresponding to the replicate ($r$) and the color to the condition ($c$). The shaded area represents output-scale values larger than the $95^{th}$ percentile, and the dotted line is the $99^{th}$ percentile. (B) Replicates with associated $\sigma^2_{f_{cr}}$ above the $95^{th}$ percentile are likely to correspond to replicates presenting either one to multiple outlier observations. **(C-D) Area Between Curves as a new metric** To replace the previously used $\Delta T_m$ as measure of the discrepancy between the fitted curves, we propose a new metric, denoted the *Area Between the Curves* (ABC). The ABC can be computed by considering the median of the observations in each condition, and computing the area between these medians. As complement to this $ABC_{median}$ metric, we propose a refined computation of the ABC using the output of the HGP model, denoted by $ABC_{GPMelt}$ (see Appendix D in S1 File). (C) A protein with at least one value $\sigma^2_{f_{cr}}$ above $q75 + 1.5 \times IQR$ is defined as presenting at least one outlier replicate ($q75$ being the $75^{th}$ percentile, and IQR the interquartile range). Comparing the differences in $ABC$ estimated using either $ABC_{median}$ or $ABC_{GPMelt}$, shows that $ABC_{median}$ likely overestimate the $ABC$ for proteins presenting at least one outlier replicate (Wilcoxon signed-rank test). (D) $ABC_{GPMelt}$ as a valid metric to replace $\Delta T_m$: considering the solubility effects reported by Sridharan et al [19], a positive, resp. negative, $ABC_{GPMelt}$ is correctly computed for desolubilized, resp. solubilized, proteins (Wilcoxon signed-rank test). **(E-F) Introduction of a new scaling factor**. The broadly used Fold Change, in which intensities in a replicate are scaled to the intensity at the lowest temperature, is compared to a newly proposed scaling, named the mean scaling. This scaling consists in scaling intensities in a replicate to the mean intensity of this replicate. (E) Scaling comparison. Considering the differences in scaled abundances at each temperature between replicates of a condition, the x-axis represents the median difference and the y-axis the variance of the differences. Results are divided by condition (control and treatment) and by scaling (fold change vs mean scaling). The panels are split in four, with the left bottom corner corresponding to reproducible observations between replicates of a condition. The three other panels reveals a lack of reproducibility between replicates. (F) Examples of proteins falling outside of the left bottom corners in panel (E). For these proteins, the results of the mean scaling and the fold change are compared.

*Between the Curves* (ABC). This measure is valid for any melting curve shapes and any number of intersection points between the curves. The thermal stabilisation (destabilisation) effect is thus only defined for cases where one melting curve remains uniformly above the other one.

As a first approximation, the *ABC* can be directly computed from the median observations (see Appendix D in S1 File, Eq (46)). Moreover, we suggest that a more precise measure of this *ABC* can be obtained using the posterior mean of the predictive distribution of the GP, as detailed in Appendix D in S1 File, Eq (49). The reason is as follows: the HGP model, which

better handles outlier observations, provides a more robust estimation of the condition-specific melting curves, and hence of the *ABC*.

Fig 6C illustrates this point. Considering the Staurosporine 2014 dataset, a protein with at least one value $\sigma^2_{f_{cr}}$ above $q75 + 1.5 \times IQR$ is defined as "presenting at least one outlier replicate" ($q75$ being the $75^{th}$ percentile, and IQR the interquartile range). We further compare the estimation of *ABC* using either the median approximation ($ABC_{median}$) or the predictive distribution obtained from fitting the HGP model ($ABC_{GPMelt}$). This boxplot shows that larger values of *ABC* are typically estimated via $ABC_{median}$ for proteins presenting at least one outlier replicate (Wilcoxon signed-rank test). Indeed, when only two replicates for each condition are available, outlier observations/replicates especially impact the median observations for this replicate, leading to over-estimated $ABC_{median}$.

Fig 6D, using the ATP 2019 dataset, further illustrates that *ABC* has a similar interpretation than $\Delta T_m$. A positive value of *ABC* corresponds to a treatment curve mostly above the control curve. More precisely, considering the effect of ATP on proteins (as reported by Sridharan et al [19]), we expect a positive *ABC* for ATP-desolubilised proteins. Indeed, the ATP-desolubilisation of proteins involves more proteins being retained into macromolecular condensates, or the macromolecular condensates to be thermo-stable at higher temperatures, thus leading to the ATP-treated melting curve to remain longer above the vehicle melting curve.

Finally, when only considering the effect size and not the direction of the changes, the *absolute ABC* might be useful. The absolute ABC, denoted $|ABC|_{GPMelt}$, consists in computing the absolute area between the curve and is strictly positive (see Appendix D in S1 File, Eq (50)). Especially, proteins successively subjected to stabilisation and destabilisation effects of approximately the same amplitude present *ABC* values close to 0, but large *absolute ABC*. This is illustrated in Fig T in S1 File.

**Scaling factor.**   With this section, we would like to introduce a new scaling of the observations to replace the broadly used Fold Change scaling. The Fold Change scaling (FC) of the observations consists in scaling intensities of a replicate by the intensity at the lowest measured temperature. This FC scaling presents several advantages, and is especially useful for melting curves interpretation. However, FC scaling is not robust to measurement errors at the first temperature and strongly influences the statistical properties of the data by focussing on the first observation of all replicates of all conditions. Furthermore, the flexible HGP models introduced in this work release all requirement on melting curve shapes, and melting behaviours can now freely diverge from sigmoidal curves starting at one. In consequence, we propose a new scaling of the observations, that we denote by *mean scaling*. It consists in scaling all observations in a replicate by the mean intensity computed across temperatures of this replicate. This scaling is more robust to measurement errors that could occur at any temperature, and allow replicate curves to start at different values. In addition to dealing with other limitations of the Fold Change scaling, detailed in Supporting Information E in S1 File, the mean scaling of the observations also improves the reproducibility between replicates. To show this, we considered the ATP 2019 dataset [19], and computed for each protein and each condition separately, the between-replicate differences in scaled observations at each temperature, for both the FC and the mean scaling. Consequently, we computed the median difference and the variance in differences for each protein and each condition. Mathematically, given a protein $p$, and considering the replicates $r \in \{1, 2\}$, the conditions $c \in \{Control, Treatment\}$, and the scaling factors $\rho_{cr} \in \{\rho_{cr}^{FC}, \rho_{cr}^{Mean}\}$, the between-replicate differences at $t_i$ for $i \in [\![2, N]\!]$ is given by:

$$\delta_{t_i}^c = \frac{\gamma_i^{c1}}{\rho_{c1}} - \frac{\gamma_i^{c2}}{\rho_{c2}} \tag{16}$$

The temperature $t_1$ has been excluded from this analysis, given that $\delta_{t_1}^c \equiv 0 \, \forall \, c$ for the FC scaling. The median difference $M^c$ and the variance $V^c$ in differences for condition $c$ are given by:

$$M^c = median(\{\delta_{t_i}^c\}_i) \quad \text{and} \quad V^c = Var(\{\delta_{t_i}^c\}_i) \tag{17}$$

We present the results of this analysis in Fig 6E. The x-axis represents the median difference $M$ and the y-axis the variance of the differences $V$. Panels correspond to the conditions (control and treatment) and scalings (fold change vs mean scaling). The panels are visually split in four using dotted lines. A condition $c$ presenting a large median difference $M^c$ and a low variance $V^c$ is expected to have two consistently different replicates. These cases will be located in the right bottom corner of each panel, and are illustrated in Fig 6F (e.g. CRELD2, WRN). Cases located in the upper left corner of each panel have a small $M$ but large $V$: we expect observations between replicates to be inconsistently different from each other, with few large differences between replicates(e.g. RPL12, treatment + FC). The upper right corner corresponds to conditions for which the differences between replicates can become very large, potentially affecting numerous temperatures (e.g. RPL12, control + FC). Reproducible observations will be located on the left bottom corner of these panels. With this plot, we show that the mean scaling concentrate most of the points in the left bottom corner, with only few conditions showing very non-reproducible replicates (e.g control condition for WRN and DST). Moreover, it can be noticed that for these cases, the FC scaling present similar differences between replicates. However, non-reproducible replicates linked to the FC scaling (e.g. CRELD2 and RPL12) are rescued by the use of the mean scaling. Focusing on CRELD2, it becomes clear that the observations at the first temperature in both conditions are outlier observations. Indeed, when comparing with the mean scaled version of these observations, only the observations at the first temperature stand out. Overall, the mean scaling performs better at maximising the similarity between replicates, being less sensitive to outlier observations, especially if the outlier is observed at the lowest temperature. Additional results and discussion can be found in Supporting Information E and in Figs U and V in S1 File.

While largely improving the statistical properties of the data, we are conscious that the mean scaling of raw intensities leads to melting curves whose shapes can appear very different from melting curves using fold changes. TPP experts might need to familiarize themselves with this new scaling and the resulting melting curve shape interpretations.

## Discussion

We presented GPMelt, a new statistical framework to analyse Temperature-Range Thermal Proteome Profiling (TPP-TR) datasets, valid on both peptide- and protein-level observations. This framework is based on hierarchical Gaussian process (HGP) models combined with a hypothesis testing framework using a non-standard statistic $\Lambda$.

This framework has been developed with the goal to overcome the main limitation of the two state-of-the-art statistical methods used in the field, namely the $T_m$ analysis [1] and NPARC [6]. Both methods, by assuming a sigmoidal shape on the melting behaviours, restrict the analysis of proteins presenting non-conventional meting behaviours. However, these proteins have received increasing attention in recent years, whether for the engendered biological hypotheses [17, 19] or for the technical challenges linked to their analysis [17, 18]. More specifically, Sridharan et al. [19] hypothesized that proteins with non-sigmoidal melting curves could undergo temperature-dependent phase transitioning. Fang et al [17] further suggested that the binding to RNA and some PTMs (especially phosphorylation and acetylation) could impact proteins melting behaviour in a non-conventional way. Pioneering the use of Gaussian

processes to incorporate non-sigmoidal melting curves in TPP analysis, Fang et al. [17] developed a so-called Bayesian semi-parametric model.

Noticing that this latest method could show high sensitivity to outlier observations, and did not yet incorporate the principle of multiple conditions, we adapted this method with a hierarchical framework. Our model better integrates outliers thanks to the bottom level of the hierarchy, making it possible to fit an individual model for each replicate while taking into account similarities between replicates of a condition. Moreover, the general structure of the hierarchy makes it theoretically possible to fit a model considering an infinite number of conditions, maximising the information sharing between all measurement for a given protein.

Furthermore, we illustrate the versatility of this model by exploring the potential of deeper hierarchies. More specifically, considering the framework of multi-task GP regression [26], we define a *task similarity matrix* for each level of the hierarchy. These matrices mathematically translate our a priori beliefs about observations similarities. Thanks to this construction, any further similarity information given by the experimental protocol can be theoretically added to the model, in the form of a new level in the hierarchy. We demonstrate this principle on a published peptide-level TPP-TR dataset [11] and investigate the principle of phosphorylation cross-talk [40] via comparisons of melting behaviours.

The proposed model relates to previously published methods. Certainly, this statistical framework is founded on the model introduced by Hensman et al. [25] and originally applied to gene expression time-series. Re-interpreting this model in light of the multi-task GP regression framework simplifies the translation of a biological protocol into a hierarchy. Especially, this allows to tackle complex experimental setups by thinking in terms of similarity matrices and curve smoothness (correlation matrix). Another related model is PairGP [41], likewise applied on gene expression time series and introducing GP regression to deal with multiple conditions and replicates. A main difference lies in the statistical testing framework suggested by the authors. In this work, we introduced a statistic $\Lambda$, inspired from a statistic previously shown to be an appropriate similarity measure for time series modeled by GPs [28]. We further estimate the null distribution of this statistic using a sampling method, following a similar process that Phillips et al. [29].

In recent years, an increasing interest in machine learning based methods to deal with TPP datasets has been observed. As examples, two deep-learning methods recently developed: DeepSTABp [42], aiming to predict protein melting point $T_m$, and an image-recognition based method [18] proposed to bypass the non-sigmoidality problem. GP and multi-task learning being important tools of the machine learning field, GPMelt can be seen as a bridge between statistical modeling of the data and the flexibility of machine learning approaches.

A possible limitation of this model is the lack of positivity constraints on the melting curves. Adding these constraints via a link function would have substantially increased the computational cost of the framework, while reducing the model interpretability. In practice, the constraints imposed by the hierarchical model and the possible additional constraints on the lengthscale typically minimise the probability of predicted negative values.

Importantly, while originally developed to improve TPP-TR dataset analyses generated using TMT, the versatile and general framework of GPMelt extends beyond TPP-TR analysis and could be applied to any continuous datasets by quantitative methods (e.g DIA) presenting replicates and conditions. A natural extension of GPMelt would be to model multiple conditions assignments, i.e adding the possibility to model some levels of the hierarchy by a sum of matrices. Moreover, a further adaptation of this model would be to infer the task-similarity matrix structure while fitting the model. This would make it possible, e.g. to cluster peptides replicates based on their melting behaviours, thus allowing proteoforms detection, as pioneered by Kurzawa et al. [14].

Collectively, the presented GPMelt statistical framework extends the analysis to the dark meltome of TPP-TR datasets, offering access to thousands of the previously excluded melting curves in both peptide- and protein-level datasets, thus paving the way to new biological discoveries on protein interactions, localisation and functions.

## Supporting information

**S1 File. Supplementary file.** Contains Appendix A-D, Supporting Information A-E, Supplementary Figures A-W and Tables A-E.
(PDF)

## Acknowledgments

We thank the EMBL IT Services HPC resources, and especially Jurij Pecar. Regarding the method implementation, we thank Federico Marotta, Renato Alves (EMBL Bio-IT), Florian Heyl and the AI Health Innovation Cluster, especially Magnus Wahlberg and Marcel Mück, for their invaluable support and insightful consultations. Regarding GPMelt methodology and biological applications, we extend our gratitude to Constantin Ahlmann-Eltze, Laurent Colbois, Brian Clarke, Oliver Stegle, Wolfgang Huber, Nikolaos Ignatiadis, Nils Kurzawa, Sindhuja Sridharan, Pablo Rivera and Tara Bartolec for insightful discussions. We thank Christophe Le Sueur for the help with the figures, and Isabelle Becher for granting permission to use and modify her icons.

## Author Contributions

**Conceptualization:** Cecile Le Sueur, Magnus Rattray, Mikhail Savitski.

**Formal analysis:** Cecile Le Sueur.

**Methodology:** Cecile Le Sueur, Magnus Rattray.

**Software:** Cecile Le Sueur.

**Supervision:** Magnus Rattray, Mikhail Savitski.

**Visualization:** Cecile Le Sueur.

**Writing – original draft:** Cecile Le Sueur.

**Writing – review & editing:** Cecile Le Sueur, Magnus Rattray, Mikhail Savitski.

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
