## [Decision Letter · Decision Letter 0]

13 Feb 2024

Dear Cécile Le Sueur

Thank you very much for submitting your manuscript "Hierarchical Gaussian process models explore the dark meltome of thermal proteome profiling experiments." for consideration at PLOS Computational Biology. As with all papers reviewed by the journal, your manuscript was reviewed by members of the editorial board and by several independent reviewers. The reviewers appreciated the attention to an important topic. Based on the reviews, we are likely to accept this manuscript for publication, providing that you modify the manuscript according to the review recommendations. As you will see the reviewer raised only minor comments and I agree with them that addressing these will improve the manuscript. We would like to apologise for the delay in getting this manuscript reviewed, a number of reviewers were unavailable over the Christmas break and early New Year grant season.

Sincerely,

Oliver Crook

Guest Editor

PLOS Computational Biology

Jian Ma

Section Editor

PLOS Computational Biology

**Guest Editor comments to authors:**

The manuscript clearly demonstrate mastery of some challenging statistical concepts, however, given the denseness of the mathematical components of the model it would be helpful to increase the accesibility of the text to those who may use the model rather than fully comprehend all the details. I suggest the authors include a high-level description of the method early on the methods sections and a summary of the main algorithmic steps at the end of the methods section.

It would be helpful to consider the following suggestions:

The advancements over ref. [17] are evident but it is not clear that this prior framework doesn't already overcome the sigmoid assumption. Appreciating the computational cost is perhaps prohibitive, it may be worthwhile demonstrating the differences empirically on a few select examples? 

It is unclear whether the p-values arising from the authors model are well-calibrated, since the authors make some remarks about being careful with p-value distribution it would be good to show empirical FDR versus observed FDR for some example especially as the heirarchy gets deepers. If the authors are struggling for a while defined ground truth it may help to generate known differences of different AUC sizes and resample residuals according to an appropriate null and alternative models. It would be useful to for user to understand how much they can trust the p-value as a measure of uncertainty quantification in this way.

Reviewer's Responses to Questions

**Comments to the Authors:**

Reviewer #1: In this paper, the authors used a hierarchical Gaussian process model to study variations in the melting profiles of proteins resulting from different treatments and multiple replications. The proposed nonparametric model, adapted from Hensman et al. 2013, does not impose any shape constraints on the melting curves. This modelling choice improves the flexibility of the model, enabling it to capture non-conventional patterns. The research is presented clearly, and the results are promising. However, I have a few questions that should be addressed prior to publication.

1. The proposed method does not impose positivity constraints on the hierarchical prior of the melting curves. As a result, not all elements in the parameter space have biologically sensible interpretations (e.g., the 95% credible band of a melting curve may include negative intensities). How should users interpret the results in such cases, and is this a concern in real-world applications?

2. Likelihood ratio test and B-H correction are used in testing the effect of different conditions. Please explain better the motivation of the use of LR test over other selection criteria such as e.g. BIC.

3. Does the posterior inference of mean functions associated with different layers (e.g. $h, g, f$,…) suffer from identifiability issues as the depth of hierarchies increases?

4. How does the proposed method scale with the size of the dataset? It would be helpful if the authors could provide relevant metrics such as wall clock time or number of function calls.

Reviewer #2: Le Sueur et al. present an important update to the analytical pipelines for TPP. TPP is already an invaluable helpful technology for an unbiased analysis of drug targets. Hence an extension to the analytical toolkit that expands the number of proteins that can be statically evaluated within an experiment would of value to the community, especially if it can scale to multiple conditions as experiments with multiple concentrations, cell lines and temperature points become more commonplace. The outlier control within the model is also a valuable addition, as having one malfunctional biological replicate could considerably compromise the results.

Important points:

• It is not clear how to obtain, download or install GPMelt and the readme file doesn’t not provide any guidance or assistance. Considering the main point of the manuscript is a tool for the community, this seems a major issue.

• The solution is presented as based on TMT where this technology is consistently losing ground to label free DIA approaches. Does GPMelt work with label free DIA and if not this would be an important method to maximise its value.

• Related to the approximate ROC curve. How likely is the treatment to inhibit all potential targets at the concentration? This is very likely to be uneven and impossible to predict, therefore this measure seems quite optimistic more than a realistic analysis. (line 385 onwards)

• Authors should clearly acknowledge the limitation of determining the specific phosphorylation site when using bottom-up proteomics for their site specific GPMelt analysis.

• Mean intensities are subject to bigger effects, especially in low replicate counts as expected in TPP experiments, with the median being far more robust why was this not used?

o Also by normalising all values by the mean, the differences between control and treatments are artificially being compressed. Only relative measure will be compared when large absolute differences could be masked.

Minor comments:

• Font sizes in figure 1 are not too small, especially the equations.

• Figure 3 graph legend font sizes are far too small to read.

• Figure 4 requires much larger font size on the legends, having too many horizontal panels makes it compressed and hard to read and interpret.

• Figure 4 Panel b should be made clear on the figure itself that the significance thresholds shown on the venn diagrams for GPMelt and NPARC are different.

• There are no comments related to the background selection for the enrichment analysis, which is what authors justify to say their method finds appropriate novel results. If no background is selected, then the enrichment results would be expected. Did the author select the correct background (the protein identified in the experiment)?

• Supplemental figures are barely visible at 200x zoom. Please make the legend clear to readers can easily identify what each colour on the plots represents, this is particularly relevant for S3 and S4 as its not possible to discern what is described on the text from the current figures.

**Have the authors made all data and (if applicable) computational code underlying the findings in their manuscript fully available?**

Reviewer #1: Yes

Reviewer #2: Yes

PLOS authors have the option to publish the peer review history of their article (what does this mean?). If published, this will include your full peer review and any attached files.

Reviewer #1: No

Reviewer #2: No

Figure Files:

Data Requirements:

Reproducibility:

References:

---

## [Decision Letter · Decision Letter 1]

23 Aug 2024

Dear Cecile Le Sueur

We are pleased to inform you that your manuscript 'Hierarchical Gaussian process models explore the dark meltome of thermal proteome profiling experiments.' has been provisionally accepted for publication in PLOS Computational Biology.

Best regards,

Oliver Crook

Guest Editor

PLOS Computational Biology

Jian Ma

Section Editor

PLOS Computational Biology

I recommend the manuscript for publication, please note the formatting issues in the supplementary material.

Reviewer's Responses to Questions

**Comments to the Authors:**

Reviewer #1: The authors have addressed most of my questions and concern. The paper is ready for publication.

Reviewer #2: The authors have addressed my comments. I would simply like to point out issues in their supplemental data. They refer to supplemental figures as S1 Fig. and later Fig. S1. some consistency would be good. Also their supplemental figures are shifted out of place, with some figures being displayed in the middle of the references section. These are minor formatting issues though.

**Have the authors made all data and (if applicable) computational code underlying the findings in their manuscript fully available?**

Reviewer #1: Yes

Reviewer #2: Yes

PLOS authors have the option to publish the peer review history of their article (what does this mean?). If published, this will include your full peer review and any attached files.

Reviewer #1: **Yes: **Hanwen Xing

Reviewer #2: No

---

## [Editor Report · Acceptance letter]

23 Sep 2024

PCOMPBIOL-D-23-01725R1 

GPMelt: a hierarchical Gaussian process models explore the dark meltome of thermal proteome profiling experiments.

Dear Dr Le Sueur,

I am pleased to inform you that your manuscript has been formally accepted for publication in PLOS Computational Biology. Your manuscript is now with our production department and you will be notified of the publication date in due course.

With kind regards,

Anita Estes
